# Adenovirus phagocytosis by neutrophils triggers a pro-inflammatory response

Salomé Laurans[1,2], Soizic Huerre[1,2], Olivier Dellis[3], Céline Férard[1], Hadrien Jalaber[1], Clément Vanbergue[2], Emilie Brun[1], Rémy Jelin[4], Oliver Nüsse[1], Karim Benihoud[2]*, Sophie Dupré-Crochet[1¤]*

**1** UMR 8000 Institut de Chimie Physique, Université Paris Saclay, Orsay, France, **2** UMR 9018 Aspects métaboliques et systémiques de l'oncogénèse pour de nouvelles approches thérapeutiques (METSY), CNRS, Université Paris-Saclay, Institut Gustave Roussy, Villejuif, France, **3** UMR 1193, Hepareg (Physiopathogenesis and treatment of liver diseases), Université Paris Saclay, Orsay, France, **4** Bioinformatics Core Facility, INSERM US23 CNRS UMR 3655, Gustave Roussy Cancer Campus, Villejuif, France

¤ Present address: Laboratoire de Génétique et de Biologie Cellulaire (LGBC), Université de Versailles Saint Quentin(UVSQ), Université Paris Saclay, Montigny-Le-Bretonneux, France
* sophie.dupre-crochet@uvsq.fr (SDC); karim.benihoud@gustaveroussy.fr (KB)

## Abstract

Adenoviruses are common pathogens that have been engineered and used for medical purposes. While their recognition by innate immune cells such as macrophages and dendritic cells is well characterized, interactions with neutrophils remain poorly understood. Using cytometry,confocal and electron microscopy, we showed that neutrophils bind to antibody-coated adenoviruses and engulf them in a phagosome. Single-cell transcriptomic approach reveals that adenovirus phagocytosis activates a specific transcriptional program in neutrophils. It also triggers calcium entry, reactive oxygen species production in the phagosome and CXCL8 release. Moreover, 4 hours after adenovirus incubation, 50% of neutrophils undergo calcium- and RIPK3-dependent cell death, accompanied by Neutrophil Extracellular Trap emission. This rapid cell death impaired complete viral degradation after 3 hours, allowing residual adenoviruses to retain genomic expression potential in target cells Thus, our data suggest that, during adenoviral infection, the neutrophil response may promote a pro-inflammatory environment that could damage host tissues.

## Author summary

Adenoviruses are common viruses that trigger mild infections, although they can lead to more severe diseases such as gastroenteritis or respiratory tract infections in immunocompromised patients. Adenovirus-derived vectors are widely used in medicine for vaccination and antitumor therapy. In this study, we investigated how adenoviruses affect neutrophils, the first immune cells recruited to the site of infection and crucial players in the immune system. We used various

**Data availability statement:** The data are within the manuscript and its Supporting Information files except for the single cell RNA sequencing. These single cell RNA sequencing data that support the findings of this study are publicly available from Gene Expression Omnibus (GEO) with the accession number GSE325281 https://www.ncbi.nlm.nih.gov/geo/query/acc.cgi?acc=GSE325281.

**Funding:** This work has been supported by a funding provided to SD-C and OD by the Graduate School Life Sciences and Health of the Université Paris Saclay (https://www.graduate-school-life-sciences-and-health-par-is-saclay.fr;France 2030 program "ANR-11-IDEX-0003). SL was a recipient of a PhD grant from Université Paris Saclay (https://www.universite-paris-saclay.fr) and also benefited for the end of her PhD from a grant from the Société Française d'Hématologie (https://sfh.hematologie.net; French program « Subventions de fin de thèse et de soudure », UPSaclay 2023-3833 convention) and from funding by Life Sciences and Health Graduate School grant. CV received a PhD grant from Université Paris Saclay and an additional fellowship from Graduate School Life Sciences and Health of Université Paris Saclay. The funders had no role in study design, data collection and analysis, decision to publish, or preparation of the manuscript.

**Competing interests:** The authors have declared that no competing interests exist.

methods to explore this interaction. We found that adenoviruses bind to neutrophils, and that this binding is enhanced by human serum containing antibodies against the viruses. This binding is followed by adenovirus internalization, which triggers several responses including the activation of specific genes, the entry of calcium into the cells, the production of reactive oxygen species (molecules that fight infections), and the release of signaling molecules to attract other immune cells. However, this process also leads to rapid neutrophil death and the release of structures called Neutrophil Extracellular Traps. Because of this quick death, adenoviruses are not fully destroyed and can still infect other cells. We suggest that this neutrophil response might create an inflammatory environment that could potentially harm the body's tissues.

## Introduction

Adenoviruses (Ads) are non-enveloped DNA viruses, containing a linear double-strand genome. Human Ads can trigger diseases such as gastroenteritis or respiratory tract infections. They often have low clinical relevance but can induce morbidity in immuno-compromised patients. Moreover, some serotypes can be dangerous as shown by recent gastroenteritis or hepatitis epidemics [1,2]. Ads are still "the most widely used vectors in the clinical gene therapy field" [3]. Hence, Ad-derived, non-replicative vectors can be used as vaccine platforms while replication-competent Ads, also known as conditionally-replicative Ads, are mainly dedicated to antitumor therapy [4,5]. Ad immunogenicity facilitates the induction of a robust immune response against the antigen targeted for vaccination and immune infiltration in tumors. In both contexts, a host immune response is set up in response to Ad interaction. Innate immunity includes numerous soluble molecules such as defensins α and β or complement proteins. Defensins can disrupt the Ad multiplication cycle in target cells [6] while complement proteins opsonize them, thus facilitating their recognition by innate immune cells. Infected cells can also participate in the anti-Ad immune response: they release proinflammatory chemokines (CXCL8, CCL1/2/3) and cytokines (TNF-α, IL-1α and β and IL-6) [7,8], amplifying immune cell recruitment and activation. Then, once innate immune cells reach the infection site, Ads can interact with macrophages or with dendritic cells (DCs) via specific receptors such as MARCO/SR-A6 (alveolar macrophages) [9] or via Fc receptors (immune receptors involved in antibody-dependent recognition) [10]. In a final step, Ad internalization by macrophages and DCs can induce lytic cell death, inflammasome activation as well as the release of pro-inflammatory cytokines [10,11].

Polymorphonuclear Neutrophils (PMNs) also contribute to the innate immune system. They play an essential role in bacterial and fungal diseases but their implication in viral infections is poorly understood. On the one hand, they are necessary for the resolution of some infections, for example Influenza A virus or Herpes simplex virus-1 [12] but, at the same time, their massive recruitment and activation lead to host tissue damage, exacerbating the disease severity, as in Covid-19 [13–15]. Another example

has been described in patients with Ad-induced severe pneumonia [16] were a high neutrophil-to-lymphocyte ratio was detected. Few studies have been performed on Ad-PMN interactions. Systemic administration of Ad vectors in mice has been associated with hepatic damage due to excessive PMN recruitment to the liver [7]. Another study focused on the molecular mechanisms behind Ad vector-PMN interaction *in vitro* and showed that Ad5, when antibody-coated, is internalized due to the action of the type 1 complement receptor (CR1) and Fc receptors (FcγRI, FcγRII, and FcγRIII) [17].

In the present study, we wanted to go further and determine *in vitro* the Ad intracellular traffic in the PMN and the subsequent consequences for both phagocytes and viruses. We also sought to discover the molecular actors and signaling pathways involved. For this purpose, we used two human Ad serotypes, Ad5 and Ad3, chosen because they belong to different Ad species having different intracellular trafficking [18]. Ad5 is the most prevalent serotype; 80% of the worldwide population has antibodies against Ad5, while Ad3 is less prevalent [19]. Our experiments were performed using two cell types: a neutrophil-like model cell line, PLB-985 cells, and human blood-purified PMNs. The results described below demonstrate that PMNs bind Ads in an immunoglobulin G (IgG)-dependent but complement-independent manner. IgG-Ad complexes were recognized by FcγRIIA receptors, then internalized in a phagosome. Subsequently, PMNs created a pro-inflammatory environment: they died rapidly, releasing Neutrophil Extracellular Traps (NETs), the pro-inflammatory chemoattractant CXCL8 and reactive oxygen species (ROS). Together, these results shed new light on Ad-PMN interactions, improving our understanding of the PMN role in viral infections.

## Results

### Ad binding to PMNs requires both IgG opsonization and the FcγRIIA receptors

To determine how PMNs interact *in vitro* with Ads, Alexa 488-labeled Ads were incubated in different conditions and then exposed to PMNs (Fig 1A-1B) or PLB-985 cells (S1 Fig). A faint association of Ads to PMNs occurred without serum, with Ad3 displaying an increased binding compared to Ad5 (Fig 1A). However, Ads incubated with pooled whole human serum (HS) showed the highest association index (measured as the mean fluorescence intensity multiplied by the percentage of Alexa 488-labeled Ad positive cells) (Fig 1B). This binding was not affected when serum was heat-treated (HI-HS), ruling out a role of the complement system (Figs 1B and S1). As we wanted to analyse the overall contribution of anti-Ad antibodies (Abs) in Ad binding, the levels of anti-Ad5 and anti-Ad3 Abs in the HS used were first checked. Our data indicated that the Ab titers were the same for both serotypes (S2 Fig). Remarkably, immunoglobulin G (IgG) depletion severely reduced Ad binding to both PMNs (Fig 1B) and PLB-985 cells (S1 Fig). Of note, with PMN, the Ad5 association index (162 000 Relative Fluorescent Units, RFU) was higher than the Ad3 association index (67 000 RFU) (Fig 1B). The percentage of Ad-associated cells was the same for PMNs (S3 Fig), around 55%, but in both, the mean fluorescence appeared higher with Ad5 than with Ad3 (200 000 RFU vs 100 000 RFU, respectively). Since the mean fluorescence reflects the number of Ads bound per cell, this difference might be due to a decrease in the number of IgG-Ad3 complexes per cell or to a difference in the efficiency of A-488 labeling. Because Ad-binding depends on IgG recognition, we sought to determine which Fc receptor was responsible for this binding. Two Fcγ receptors, FcγRIIA (CD32) and FcγRIIIB (CD16) are constitutively expressed on PMNs. To determine the respective role of these receptors in Ad binding, we incubated PLB-985 cells with CD16 or CD32-blocking antibodies [20–22] before adding Ad5 or Ad3 pre-incubated with HS. Ad binding to PLB-985 cells was disturbed mainly when the CD32 was blocked (Ad5, Ad3; Fig 1C and 1D).

### Ads are entrapped in phagosomes

Having shown that Ad binding to PMNs depends on both IgG and CD32/FcγIIA receptor, we examined whether this binding was followed by internalization. Confocal microscopy using A-568-phalloidin labeling of PMNs and PLB-985 cells showed that HS-opsonized Ad5 and Ad3 were internalized inside PMNs over time (Fig 2A). Going further, transmission electron microscopy (TEM) used to examine PMN and PLB-985 cell membranes confirmed that Ad5 were internalized

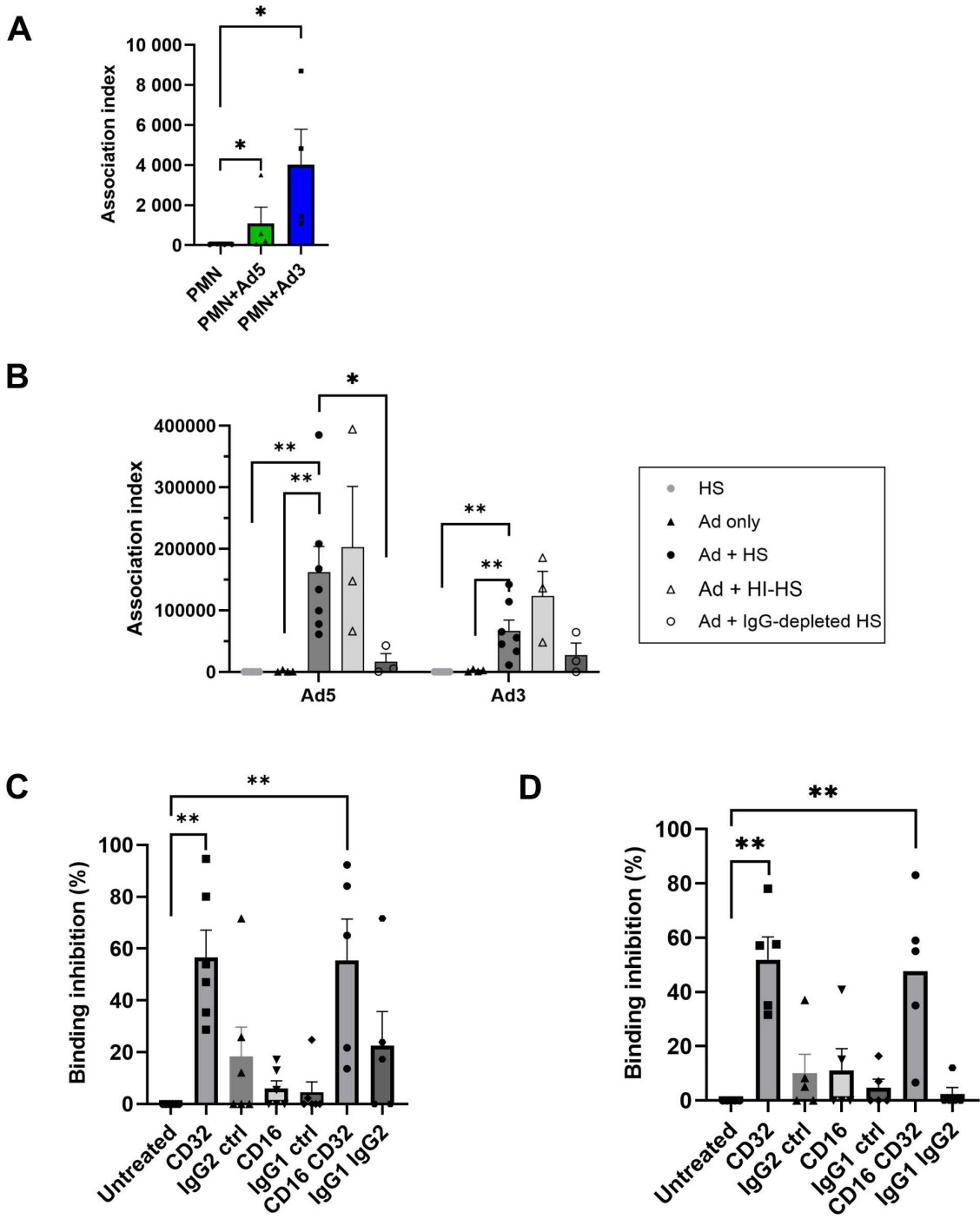

**Fig 1. IgG-dependent Adenovirus 5 (Ad5) and Ad3 binding to polymorphonuclear neutrophils (PMNs) & PLB-985 cells. (A, B)** Alexa-488 (A-488) labeled Ad5 or Ad3 were incubated at 37°C with or without human serum (HS), heat-inactivated (HI-HS) or IgG-depleted HS. Then PMNs were exposed to Ads at 4°C to allow cell binding (MOI $10^4$ vp/cell) and analysed by flow cytometry. The results are presented as an association index + SEM (*i.e.,* mean fluorescence intensity multiplied by percentage of positive cells) (n ≥ 3). Each dot represents the value for one experiment. **(A)** Ad association index without serum. **(B)** Ad association index in the different conditions described above. Mann-Whitney tests: *, p < 0.05; **, p < 0.01. **(C, D)** A PMN model cell line, PLB-985 cells, was untreated or incubated with blocking antibodies against CD16 and/or CD32 receptors, or with control isotypes. Then cells were incubated with HS-pre-incubated A-488 Ad5 **(C)** or Ad3 **(D)** at 4°C to allow binding. Results show the percentage of binding inhibition (+ SEM) for the different Ad + HS conditions (five independent experiments) relative to untreated cells exposed to HS-opsonized Ad: Mann-Whitney test * p < 0.05; ** p < 0.01.

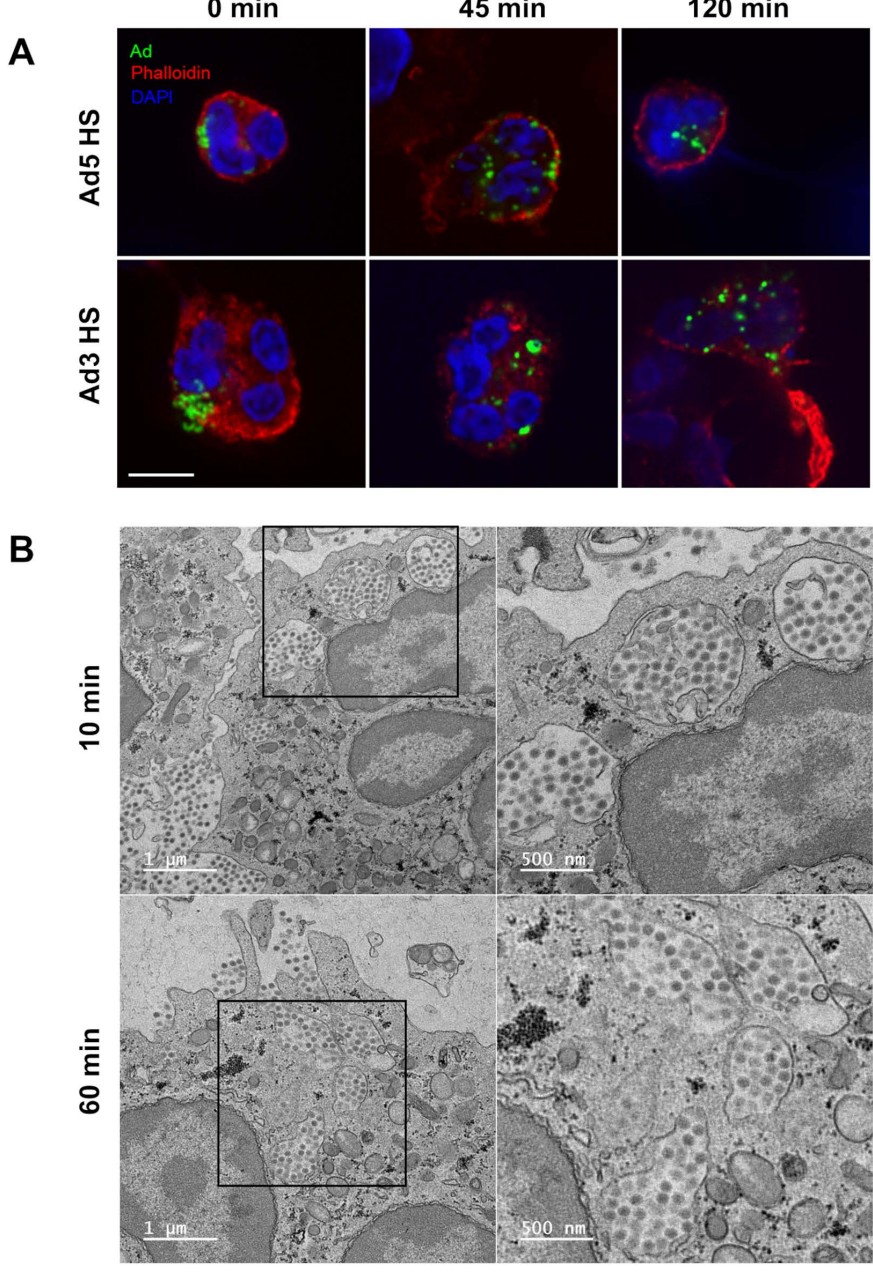

**Fig 2. Ad5 and Ad3 are internalized in PMNs as patches. (A)** PMNs were incubated for different times (0, 45 or 120 min) with HS-opsonized A-488 Ads (MOI 10⁴ vp/cell). Cortical actin network and DNA were labeled respectively with A-568 phalloidin and DAPI. Scale bar = 10 μm. The images represent a single plane from a Z-stack. **(B)** PMNs were incubated with HS-opsonized Ad5 (MOI 10⁴ vp/cell) for 10 min or 1h at 37°C, then treated for electron microscopy observation. Representative TEM images of Ad5 containing phagosomes.

in membrane-delimited compartments (Figs 2B and S4). We could not detect any Ad5 free in the cytosol after 60 min of internalization. To determine the identity of the Ad-confining vesicles and the nature of the proteins involved in membrane trafficking during internalization, a colocalization study exploring the involvement of specific markers was performed on PMNs, using confocal microscopy (Fig 3A). After 10 min incubation at 37°C, Ad5 and Ad3 colocalized with Rab5, an

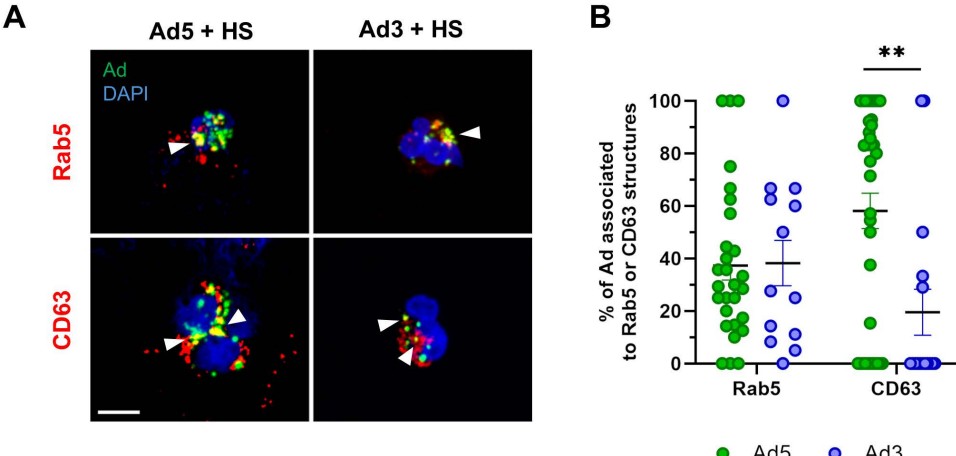

**Fig 3. Ad5 and Ad3 patches colocalize with PMN phagosomal markers. (A)** Representative images of Ad colocalization with organelles using confocal microscopy imaging. PMNs were incubated with HS-opsonized A488-Ad5 or Ad3 (MOI $10^4$ vp/cell, green dots) for 10 min, fixed, permeabilized and then immunostained against the early endo/phagosomal marker Rab5 (red) or the primary granule marker CD63 (red). DNA was stained with DAPI (blue). The arrowheads show the colocalization spots. Scale bar = 5 μm. The images are Z-projections generated from Z-stacks. **(B)** The graph represents the average percentage ± SEM of Ad patches associated with Rab5- or CD63-positive phagosomes (each dot represents the percentage of Ad patches associated with Rab5- or CD63-positive phagosomes in one cell; for Rab5: 27 (Ad5) and 13 (Ad3) positive cells were analyzed; for CD63: 39 (Ad5) and 25 (Ad3) positive cells were analyzed). The 3D colocalization analysis was performed with the ImageJ software (JaCoP plugin). Mann-Whitney test: **, p < 0.01.

early-endosome and phagosomal marker, and with the primary granule marker CD63 [23], confirming the Ad internalization in a phagosome. A difference was observed for the percentage of Ad5 and Ad3 colocalized with CD63 (58% for Ad5; 24% for Ad3) (Fig 3B) and we suggest that this could be due to less fusion of cytoplasmic primary granules with phagosomes in the case of Ad3.

### HS-opsonized Ads induce NOX2-dependent reactive oxygen species production and intracellular calcium increase

Once internalization was confirmed, we sought to determine if PMNs produced reactive oxygen species (ROS) in response to Ad encounter. HS-opsonized Ad3 or Ad5, but not "naked" Ads, were able to induce ROS production, as assessed by chemiluminescence assay (Fig 4A). Preincubation of PMNs with DPI, a NADPH oxidase inhibitor, completely abrogated ROS production elicited by HS-opsonized Ads (Fig 4A), indicating that NADPH oxidase (NOX2) was the ROS source. The involvement of this enzyme was confirmed using NOX2-deficient PLB-985 cells in which Ad exposure produced no increase in ROS, when compared with WT PLB-985 cells (S5 Fig). This ROS production could be detected inside the PMNs using the OxyBURST-BSA probe, a fluorescent ROS-sensitive probe that is internalized with the Ad in the PMNs. The ROS signal was observed in the vicinity of the Ad patches in PMNs (Fig 4B). Furthermore, p67[phox], a NOX2 cytosolic subunit, necessary for NOX2 activation, was detected surrounding the Ad patches after 10 min incubation (Fig 4C). This result corroborates our hypothesis that ROS production is due to cytosolic and membrane NOX2 subunits assembled at the Ad-positive phagosomes.

Ad interaction with PMNs also triggers an increase in cytosolic calcium concentration. More precisely Ad5 interaction with PMNs and PLB-985 cells triggered a progressive increase in cytosolic calcium over time (Figs 4D and S6). This rise was detected with Ad5 as well as Ad3 (S6A Fig). In the presence of the calcium-chelating agent EDTA (2.5mM) in the extracellular medium, no cytosolic calcium increase was observed. Thus, calcium could only be made available from intracellular stores. However, no early calcium peak was detected suggesting that $Ca^{2+}$ was not massively released from the

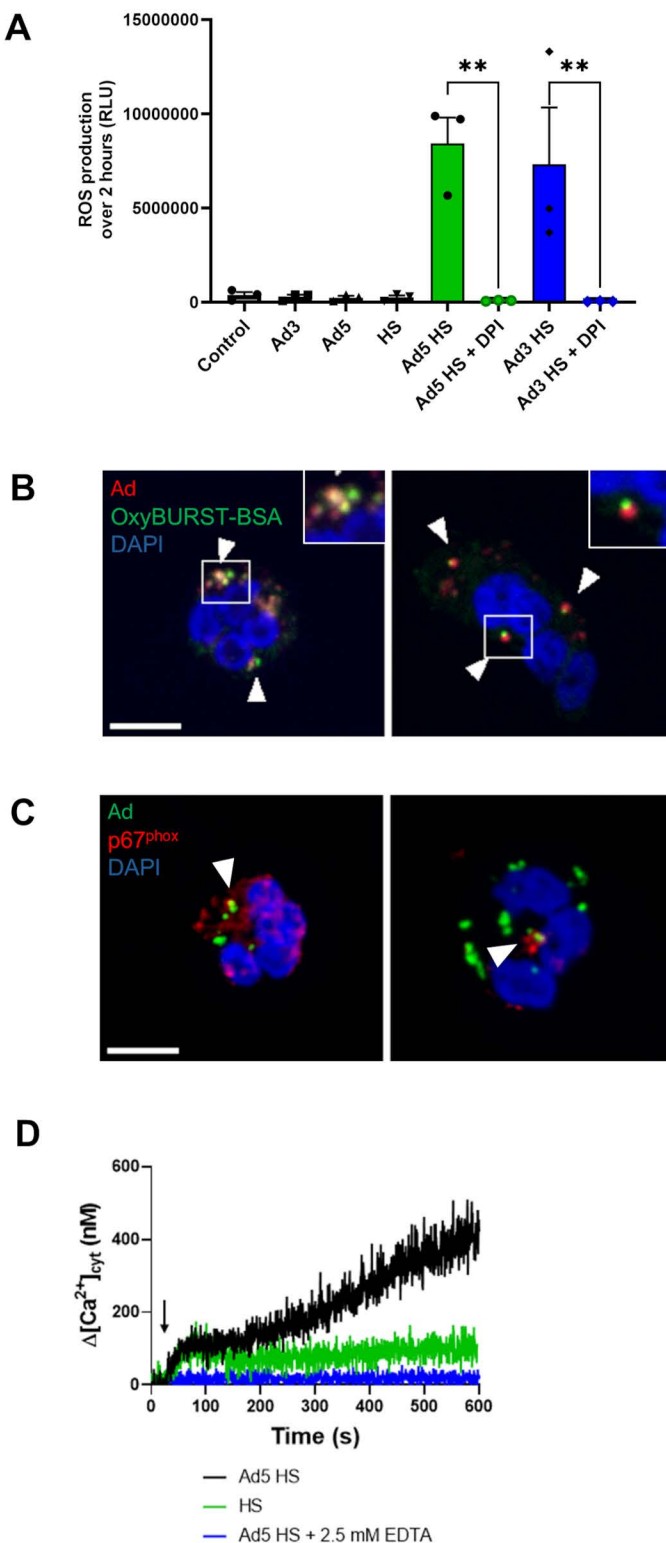

**A**

**B**

Ad
OxyBURST-BSA
DAPI

**C**

Ad
p67phox
DAPI

**D**

**Fig 4. Ad5 and Ad3 trigger NOX2-dependent ROS production and induce an extracellular calcium-dependent Ca²⁺ increase in PMNs. (A)** PMNs were incubated with buffer (control), HS, buffer-incubated Ad, or HS-opsonized Ad (MOI 10⁴ vp/cell) ± diphenyleneiodonium chloride (DPI), a NADPH oxidase inhibitor. The graph shows the integrated ROS production over 2 hours + SEM, detected by a luminometry-based test from three independent

experiments. Two technical replicates were performed for each experiment. Each dot represents the mean value of the technical replicates for each experiment. Statistical significance determined by Kruskal-Wallis test followed by Dunn's multiple comparisons tests. **, p < 0.01. **(B)** Representative images of Ad-exposed PMNs preincubated with OxyBURST-BSA probe (green). After 5 min of OxyBURST-BSA incubation, PMNs were incubated with HS-opsonized Star Orange-labeled Ads (MOI $10^4$ vp/cell) for 45 min at 37°C. Scale bar = 5 µm. Inset: 2.5x zoom of the region of interest. Arrowheads and insets show fluorescent OxyBURST-BSA surrounding Ad patches. Three independent experiments were performed. **(C)** Representative images of p67$^{phox}$ immunofluorescence. PMNs were allowed to internalize HS-opsonized A-488 Ads (MOI $10^4$ vp/cell) (green) for 10 min at 37°C then cytosolic NADPH oxidase p67$^{phox}$ (red) was immunodetected and the nuclei (blue) were stained. Arrowheads show p67$^{phox}$ signal surrounding Ad patches. Scale bar = 5 µm. **(D)** Cytosolic $Ca^{2+}$ concentration ($[Ca^{2+}]_{cyt}$) measurement in PMNs using Indo-1 fluorescence. Cells were first preincubated with either buffer alone or with buffer + EDTA 2.5 mM for 5 min. Then, after 30 s control measurement, either human serum (HS) (green trace), or HS-opsonized Ad5 (black and blue trace) were added (black arrow). Cytosolic $Ca^{2+}$ did not increase when extracellular calcium was chelated in the presence of EDTA (blue trace). The graph shows a single result, representative of 3 experiments.

intracellular calcium store. Therefore, the Ad-induced increase in cytosolic calcium is mainly due to extracellular calcium entry rather than to mobilization of the intracellular calcium store (Figs 4D and S6B). As a control, we checked that chelating calcium with EDTA didn't affect Ad phagocytosis (S7 Fig).

## Ad exposure activates a specific transcriptional program in PMNs

Several transcriptomic studies performed on PMNs showed activation of transcriptional programs specific to a danger signal or after bacterial infection [24–26]. To determine if such transcriptional changes also occur on Ad interaction, a single-cell RNA sequencing study was performed on blood purified PMNs exposed to HS alone or to HS-opsonized Ad5. The Seurat software classified cells based on their RNA expression profiles into clusters numbered from 0 to 6, which are visualized using Uniform Manifold Approximation and Projection (UMAP) (S8A Fig). Clusters 0, 1, 3, 4 and 6 expressed the genes (*NAMPT, CXCR2, SOD2, CSF3R, FCGR3B*) that defined peripheral blood PMNs as found by Montaldo *et al.* [26], although the expression was weaker for cluster 1 (S8B Fig). The differential expressions of the two non-coding RNAs *MALAT1* and *NEAT* were segregated in the previous cluster whereas the Differentially Expressed Genes (DEG) in cluster 4 were the ISGs (interferon stimulated genes) (S8B Fig). These subpopulations were previously observed in human PMNs [25,27]. Cluster 6 was specific to PMNs incubated with Ad. This Ad-specific PMN subpopulation was characterized by several DEGs including *CXCL8, G0S2, H3-3B, FTH1, SAT1* and *BCL2A1* (Fig 5A). Comparing the whole transcripts in the HS and HS-opsonized Ad5 conditions (pseudobulk approach), we showed that the genes cited above were upregulated in the HS-opsonized Ad5 condition compared to the HS condition (Fig 5B). Amongst the Ad population specific markers, the most remarkable differences of expression, compared in the 2 conditions, were noted for the G0/G1 switch gene (*G0S2*) and *CXCL8* (Fig 5A). *G0S2* has been involved in apoptosis [28] but it is also an inhibitor of adipose triglyceride lipase (ATGL) which degrades triglyceride-rich lipid droplets in PMNs [29]. Going further, RT-qPCR confirmed the *CXCL8* transcriptional increase in PLB-985 cells (Fig 5C) and an increase for both *G0S2* and *CXCL8* transcripts in PMNs exposed to HS-opsonized Ad5 (S9 Fig).

## Ads induce a NET-associated RIPK3- and calcium-dependent cell death in a few hours

Because single-cell RNA sequencing revealed upregulation of apoptosis-related genes such as *G0S2*, we wondered if Ads affected PMN cell death. The PMN lysis was observed using a Sytox Green-based cell death assay. Sytox Green is a cell-impermeant fluorescent DNA probe. Its fluorescence signal increases if cell membrane integrity is lost or if NETs are released. The Fig 6A shows that the cell death percentage increased significantly over time in the presence of both HS-opsonized Ad5 and Ad3 compared to HS-exposed PMNs: 52% and 61% of PMNs were dead after 4 h incubation with Ad5 and Ad3, respectively. PMNs can respond to infection by producing NETs: cells expel their decondensed DNA, associated with histones and granule proteins such as myeloperoxidase (MPO) into the extracellular environment to avoid infection spreading in the

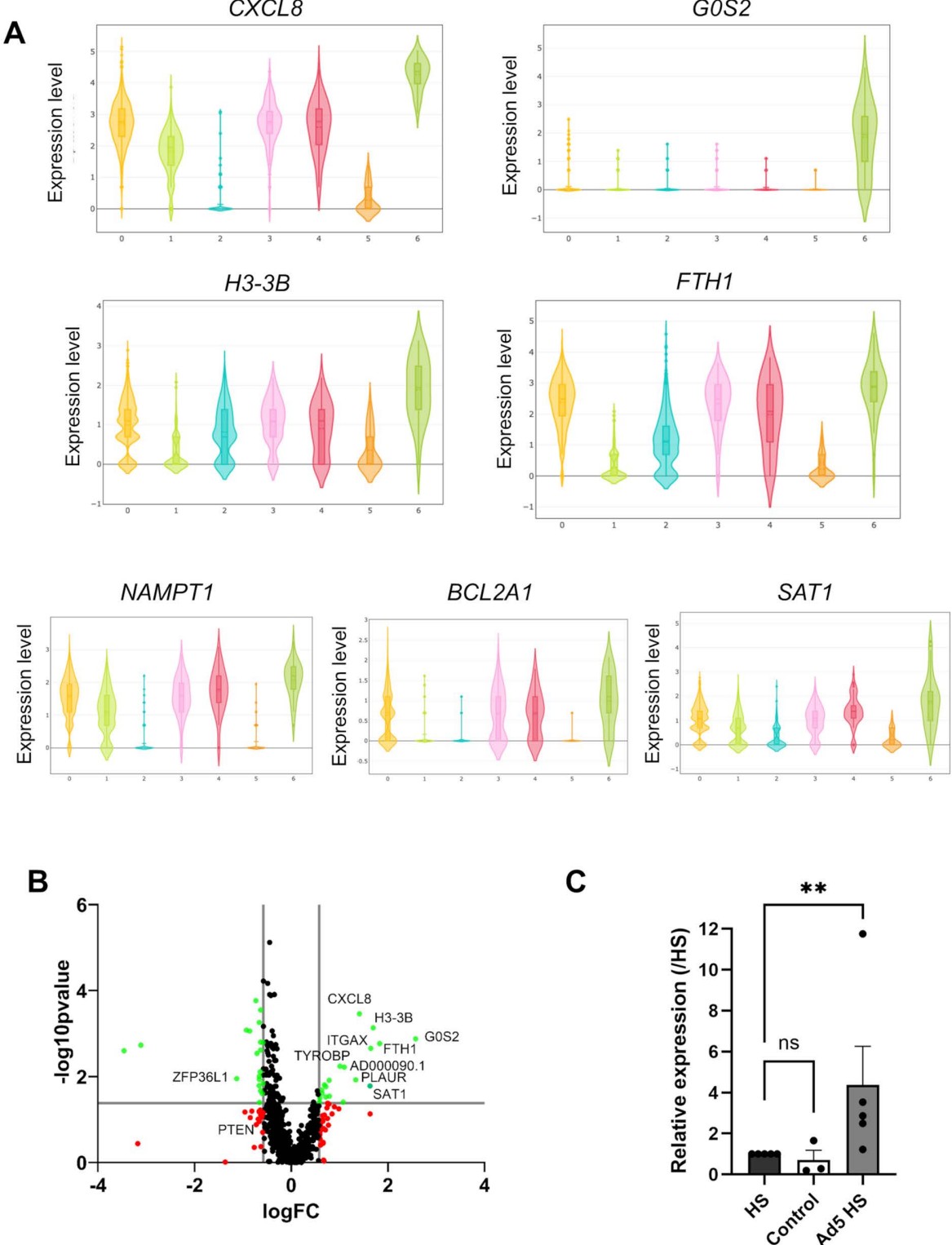

**Fig 5. Gene expression modulation in PMNs in interaction with Ad5. (A)** Single-cell RNA sequencing grouped analysis of blood purified PMNs incubated with HS or with HS-opsonized Ad5. The log-normalized expression of the most Differentially Expressed Genes in cluster 6 as analyzed by Seurat package. Cluster 6 corresponds to a specific PMN population incubated with Ad5 in the presence of HS. The graphs were generated using CerebroApp

package in RStudio. **(B)** Volcano plots comparing pseudobulk gene expression of PMNs with human serum-opsonized Ad5 vs. PMNs with human serum. The –log10-transformed p-value (Y axis) is plotted against the log2-transformed fold change (FC, X axis). P-value_Cutoff_line = 0.05, log2Fold Change_Cutoff_line = 0.58/-0,58. Black points: no significant difference; red points: log2FC above the cutoff values; green dots: log2FC & p-value show a significant difference. Volcano plot was generated using GraphPad Prism version 9. **(C)** Analysis of *CXCL8* expression in PLB-985 cells by RT-qPCR. The histogram shows fold change after normalization with *GAPDH* gene expression, in comparison to cells incubated with human serum. Data are from three (Control) and five (HS and Ad5 HS) experiments with two technical replicates for each experiment. Each dot represents the mean value + SEM of the technical replicates for each experiment. Mann-Whitney test was performed: **, $p < 0.01$; ns, non-significant.

organism. Thus, we wanted to know if Ads could trigger the formation of NETs. By immunofluorescence, we detected web-like structures positive for MPO and DAPI after 4 h of incubation with phorbol myristate acetate (PMA), a compound well-known for its capacity to trigger NETs [30]. Interestingly, as shown in Fig 6B, NETs were also observed after PMN incubation with HS-opsonized Ad5 and Ad3. S10 Fig displays various images of NETs 3 hours after PMN incubation with Ad5. The images reveal decondensed chromatin filaments associated with membrane fragments. Additionally, some Ad particles were trapped within the NETs (S10A and S10B Fig). The same results were obtained with the model cell line PLB-985 (S11 Fig).

PMN death and NET emission are hallmarks of a type of cell death called NETosis [31] but also occurs during necroptosis [32,33]. To determine more accurately how cell death was triggered, we incubated PLB-985 cells in the presence of HS-opsonized Ad5 with inhibitors targeting different cell death pathways (summarized in Fig 7A) and looked at the Sytox-positive cell percentage as compared to Ad-exposed cells treated with vehicle. Only GSK872, an inhibitor of RIPK3, a protein involved in necroptosis, induced a significant decrease in Ad5-induced Sytox-positive cell percentage after 4 h incubation (Fig 7B). In parallel, we investigated the role of some FcγR signaling pathway components. Neither Src inhibition by dasatinib nor Akt inhibition delayed Ad-induced cell death (Fig 7C). Since Ads trigger extracellular calcium entry, we examined the impact of chelating extracellular $Ca^{2+}$. Pre-incubation of PLB-985 cells with EDTA induced a decrease in Sytox-positive cell percentage (Fig 7C). Similar results were obtained for PMNs: data showed that GSK872 and EDTA decreased Ad5-induced PMN death (Fig 7D). In order to strengthen the implication of RIPK3, we used two others inhibitors: GSK 840 and GSK843. Fig 7D shows that both decreased the Sytox-positive cell percentage. We also explored whether ROS were involved in PMN death. Sytox Green assay indicated that PMNs died in a NOX2-independent manner since NOX2 inhibition by DPI had no effect on PMN death (Fig 7D). This was confirmed by Sytox Green assay performed on a NOX2-deficient PLB-985 cell line. No difference in Ad5-induced cell death was observed when comparing wild-type and NOX2-deficient cells, in contrast to PMA-induced cell death, which is NOX2 dependent (S12 Fig). The same results were also observed with PMNs exposed to opsonized Ad3 (S13 Fig).

### Ads trigger RIPK3-dependent but ROS-independent CXCL8 release

PMNs can influence the immune response via chemokine and cytokine release and our single-cell RNA-sequencing data indicated that PMNs exposed to Ads upregulated *CXCL8* gene expression. Thus, CXCL8 release was assessed after PMN incubation with HS-opsonized Ad5 or Ad3. After 1h incubation, both Ads induced significant CXCL8 release, compared to PMNs exposed to HS alone and this release increased after 3h incubation (Fig 8A). To better understand the molecular factors involved in this release, we assessed the ROS impact on CXCL8 release. ROS did not affect CXCL8 release, as shown after PMN treatment with the NOX2 inhibitor DPI (Fig 8B). In addition, no significant difference in CXCL8 release was observed between WT and NOX2-deficient PLB-985 cells after exposition to Ads (Fig 8C). Furthermore, since Ad-induced lytic PMN death in a RIPK3-dependent manner, PMNs were incubated with GSK872 prior to Ad exposure. GSK872 strongly decreased the CXCL8 ratio (*i.e.,* CXCL8 concentration in the presence of HS-opsonized Ads compared with CXCL8 concentration with HS) at 3h incubation, underlining the role of RIPK3 in CXCL8 release (Fig 8B). CXCL8 increase could be involved in a PMN self-sustaining activation loop, as seen with CXCL8 in SARS-CoV2 infection [34]. Indeed, in this study, CXCL8 enhanced PMN death by binding to its receptors (CXCR1/2). However, in our study,

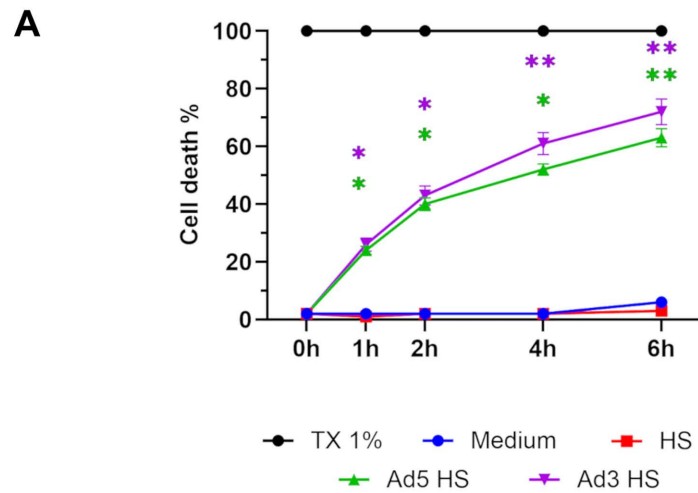

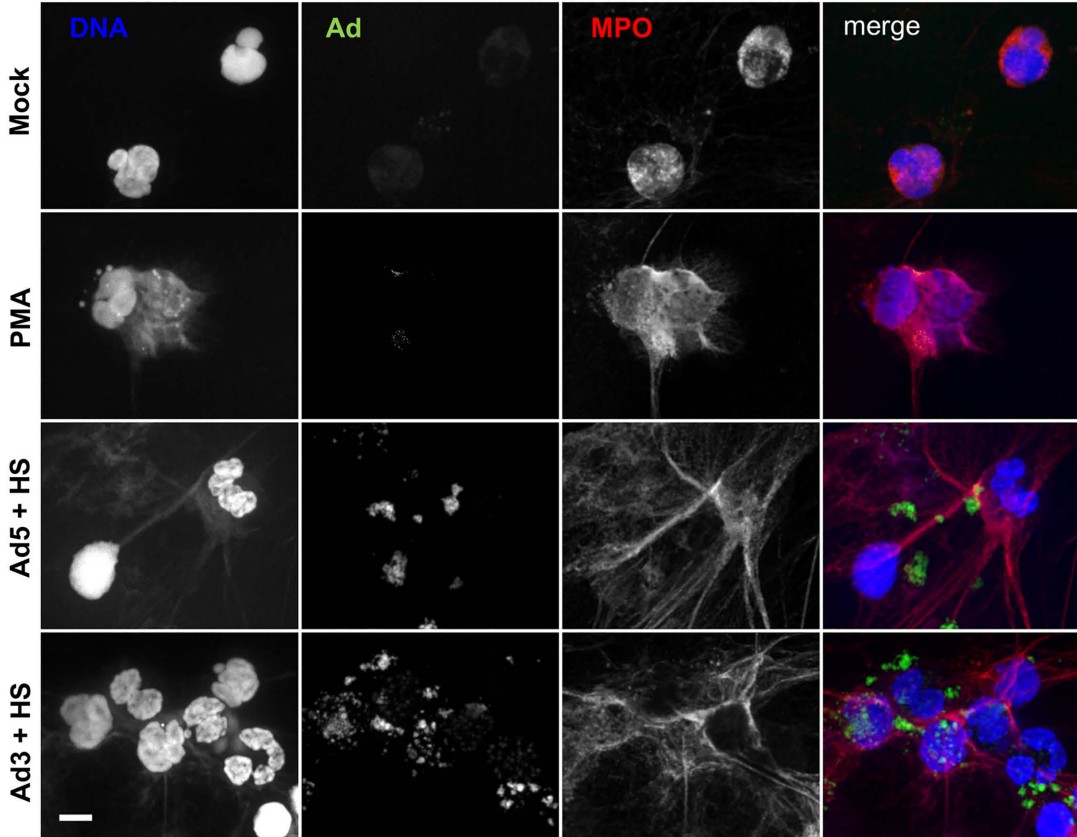

**Fig 6. Ad5 and Ad3 internalization induces PMN lysis with NET emission. (A)** Ad cytotoxicity assessed via Sytox Green assay. PMNs were incubated with HBSS, Triton-X-100, human serum (HS) or HS-opsonized Ads (MOI $10^4$ vp/cell) in the presence of the cell-impermeant fluorescent probe Sytox Green probe, a cell-impermeant DNA fluorescent probe. Cell fluorescence was measured with a plate reader. The curves represent the percentage (± SEM, 5 independent experiments) of PMN death (*i.e.,* PMN fluorescence relative to Triton-X-100-lysed PMN fluorescence multiplied by 100) at different incubation times. Kruskal-Wallis test was performed followed by Dunn's multiple comparison tests: *, p<0.05 (comparison with HS). **(B)**

Ad-induced NET (neutrophil extracellular trap) formation. PMNs were incubated with buffer (control), PMA 100 nM (positive control) or HS-opsonized A488-Ad5 or Ad3 (MOI 10⁴ vp/cell) for 4h. Then, after cell fixation, cells were permeabilized, immunolabeled for myeloperoxidase (MPO) and stained with DAPI. The images are Z-projections generated from Z-stacks. Three independent experiments were performed. Scale bar = 10 μm.

an inhibitor of the CXCL8 receptor (reparixin), failed to reduce PLB-985 cell death (Fig 7B). An explanation could be that there is not enough CXCL8 secreted to induce PMN death as it requires high receptor-saturing concentration [35].

### After PMN phagocytosis, Ads are still able to transduce target cells

As Ads trigger PMN death, we wondered if phagocytosed Ads were killed before PMN death. The Ads used in this study carried reporter genes (*lacZ* for Ad5, *gfp* for Ad3), so their functional genome was assessed by measurement of β-galactosidase activity (Ad5) or GFP expression (Ad3) in 293A and A549 target cells, respectively. PMNs were incubated with HS-opsonized Ads for 15 min or 3h. Cells were then harvested and lysed to free Ads. The lysate was added to target cells to monitor the functionality of the remaining Ad particles, assessed according to β-galactosidase activity (Ad5) or GFP fluorescence (Ad3). No difference was observed in the numbers of functional Ad particles after 15 min or 3h of phagocytosis (Fig 9). These results suggest that Ad phagocytosis by PMNs has only a limited impact on Ad degradation, at least at 3h p.i.

## Discussion

Because the role of PMNs in viral infections is still poorly understood and Ads are widespread pathogens also used as tools in medical applications, including non-replicative vectors for vaccination and oncolytic viruses for virotherapy, this study aimed to better understand the PMN-Ad encounter and its consequences, *in vitro*. Our results have shown that PMNs bind to HS-opsonized Ads and internalize them in phagosomes. This phagocytosis triggers extracellular calcium entry, ROS production and *CXCL8* transcription and release. It also induces the transcription of a specific set of genes and later cell death, which is RIPK3- and calcium-dependent and accompanied by NET emission. All these results are summarized in Fig 10. The fact that CXCL8 release is lowered by RIPK3 inhibition suggests that this process could be linked to PMN death. As a consequence of this rapid PMN death, engulfed Ads are poorly destroyed. Binding assays showed maximal association with HS-opsonized Ads: PMNs bound Ads when they were coated with IgG, and this recognition involved the FcγRIIA. Influenza A Virus (IAV) is also detected by PMNs when its fiber stacks are IgG-coated and their interaction implies a Fcγ receptor recognition [36].Moreover activation of monocyte-derived dendritic cells by Ad5 involved the FcγRIIA [37]. A faint binding was observed for both Ads in the absence of serum: Ad5 usually binds to target cells using Coxsackie and Adenovirus receptor (CAR) while Ad3 uses desmoglein-2 (DSG-2) and CD46. CAR and DSG-2 receptors are not expressed by leukocytes, unlike CD46 [38]. This might explain the slight increase in the Ad3 association index without HS. Heat inactivation of HS did not disturb Ad binding to PMNs, suggesting that complement proteins did not participate in Ad-PMN association. These results are partially consistent with the observations made by Cotter *et al.* [17] who observed an IgG- and complement-dependent binding between Ads and PMNs. Moreover, Perreau *et al.* show that activation of monocyte derived dendritic cells by Ad5, opsonized with heat inactivated pooled sera, also involved the FcγRIIA [37]. After their recognition, Ads are internalized in CD63⁺ Rab5⁺ phagosomes, as observed by confocal microscopy. Indeed, 1h after phagocytosis, phagosomes containing numerous Ads were observed by TEM. Ad escape from cell endosomes or phagosomes has been observed in epithelial cells [39,40] and in macrophage-like and monocyte-derived dendritic cells [10,41]. However, we could not detect any Ads in the cytosol in PMNs after 1 h of phagocytosis in our TEM images, suggesting that these phagosomes are not ruptured. The lack of Ad release in the cytosol could be linked to the presence of α−defensin in neutrophil phagosomes after their fusion with primary granules. Indeed, Smith *et al.* reported that α−defensin blocks Ad5-induced phagosomal rupture [6].

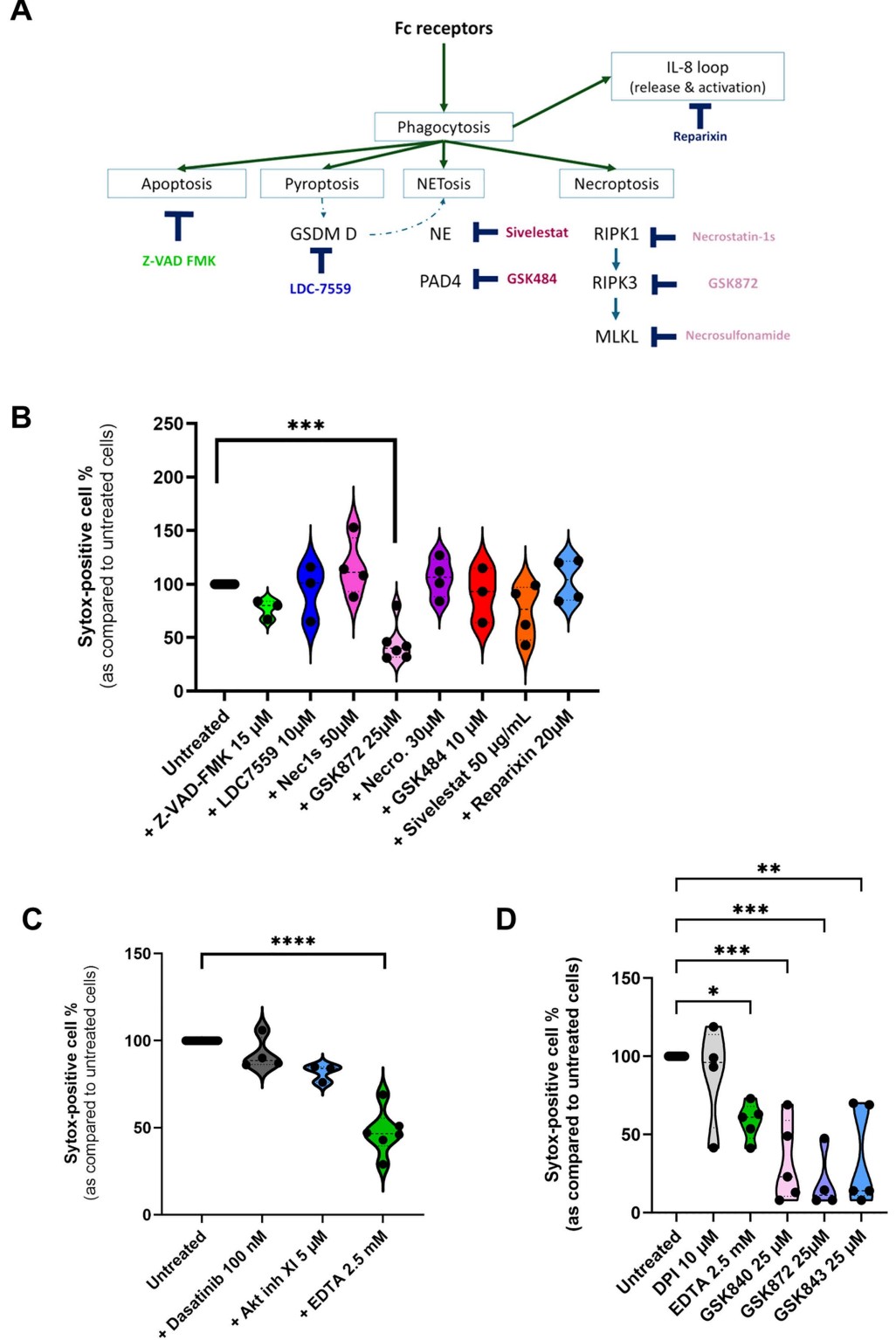

**Fig 7. RIPK3- and calcium-dependent Ad5-induced cell death. (A)** Scheme of cell death molecular actors indicating the inhibitors and their targets in the different studied pathways. **(B)** PLB-985 cells were incubated with Sytox Green probe, various cell death inhibitors and with HS-opsonized Ad5. The graph shows Sytox Green fluorescence compared to the Ad-exposed cells treated with the respective inhibitor or HS after 4h (± SEM). The condition

without inhibitors (untreated cells) was set at 100%. Kruskal-Wallis' test followed by Dunn's multiple comparison tests were performed. ***, p<0.005. **(C)** The same experiment as in (B) was performed, using various inhibitors of Fc receptor signaling. Kruskal-Wallis test was performed followed by Dunn's multiple comparison tests. **(D)** PMNs were pre-incubated with different inhibitors in the presence of Sytox Green probe and incubated or not with HS-opsonized Ad5. After 4h, Sytox Green fluorescence was measured and expressed as a ratio relative to the fluorescence of the Ad-exposed cells treated with the respective inhibitor or HS. The condition without inhibitors (untreated cells) was set at 100%. Kruskal-Wallis test was performed followed by Dunn's multiple comparison tests: **, p<0.01; ***, p<0.005. For B, C, D: three independent experiments with three technical replicates per condition for each experiment were performed.

PMN-Ad interaction led to a multitude of consequences for PMNs. Single-cell RNA sequencing identified distinct subpopulations of PMNs within blood-derived PMNs, and incubation with Ad induced the emergence of a specific PMN subpopulation. The latter was characterized by a specific gene signature including *CXCL8, H3-3B, FTH1* genes and apoptosis-related genes *BCL2A1* and *G0S2*. *CXCL8* has been shown to be upregulated in PMNs of patients with severe COVID-19 [34] and is a potent chemoattractant for PMNs [35]. B-cell lymphoma 2-related protein A1 (BCL2A1) is an anti-apoptotic protein belonging to the BCL2 protein family, a target of NF-kB [42]. Another target of NF-kB is G0S2 which has been described as a pro- or anti-apoptotic protein [28,43]. This increased expression of NF-kB target genes in this PMN subpopulation could be due to the activation of MAPK signaling pathway. G0S2 is also an inhibitor of adipose tri-glyceride lipase (ATGL) [44]. ATGL is involved in triglyceride rich lipid droplet degradation in PMNs, leading to the release of lipid mediators [29]. Lipid droplets are induced by viral infection in macrophages and are required for an interferon response against the viruses [45]. Degradation of lipid droplets has been linked to survival during PMN infection by *Pseudomonas aeruginosa* [46]. Thus, upregulation of G0S2 may lead to accumulation of lipid droplets favoring cell death and inhibiting lipid mediator release. In the present study, we did not go further in the analysis of G0S2 upregulation consequences. *G0S2*, *CXCL8* and *FTH1*, 3 characteristic genes of this Ad-PMN specific population, have also been described as signature genes of a PMN population highly predictive of septic shock [24]. Moreover, PMNs from broncho-alveolar liquid fluid of patients with Acute Respiratory Distress Syndrome have increased in *FTH1* expression [47]. Taken in context, our results add weight to the conclusion that *G0S2*, *CXCL8* and *FTH1* are part of the gene signature of this proinflammatory PMN sub-population.

An intracellular calcium concentration increase was also observed upon Ad-PMN interaction. Previous studies on PMNs have shown that a rise in cytosolic calcium is essential for full activation of NADPH oxidase during IgG-opsonized phagocytosis [48]. This increase in cytosolic calcium relies on calcium release from the endoplasmic reticulum followed by the interaction between the stromal interaction molecule 1 (STIM1) and the plasma membrane channel ORAI [49] allowing a Store-Operated Calcium Entry (SOCE). Our results indicate that Ad incubation with PMNs predominantly induces an extracellular calcium entry. However, we cannot rule out a contribution of intracellular stores. One of the consequences of Ad internalization by PMNs is NOX2-dependent ROS production. This ROS production is consistent with the PMN ROS production observed in mice infected with HSV, RSV or IAV [50–52]. Notably, Ad-induced ROS production in PMNs does not appear to affect cell death. However, ROS may modulate signaling pathways and contribute to a pro-inflammatory environment *in vivo*, which may lead to excessive recruitment and activation of PMNs at the infection site as seen with IAV, SARS-CoV2 or RSV [12,34,53].

Ad5 opsonized with pooled serum IgG triggered cell death of human monocyte-derived macrophages [54] and monocyte-derived dendritic cells [10]. Labzin *et al.* showed that this pyroptosis only occurs for Ad5 opsonized with high concentration of pooled serum IgG. We also observed that HS opsonized Ad internalization by PMNs triggered rapid and massive PMN death. This death is Src-independent as the use of dasatinib, a Src kinase inhibitor, did not prevent it. Futosi *et al.* have shown that up to 100 nM of dasatinib did not affect opsonized *S. aureus* phagocytosis by PMNs [55], which could explain its inefficiency in our experiments. After 4 h incubation with Ads, more than 50% of PMNs were dead and

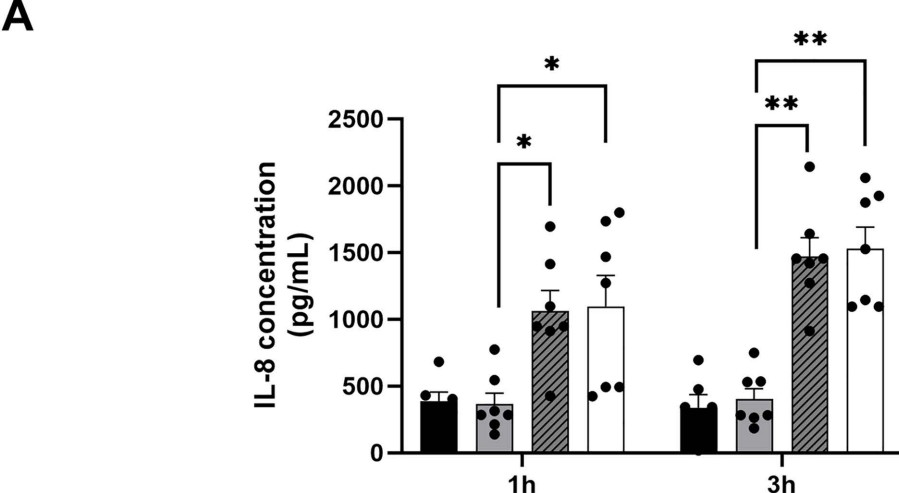

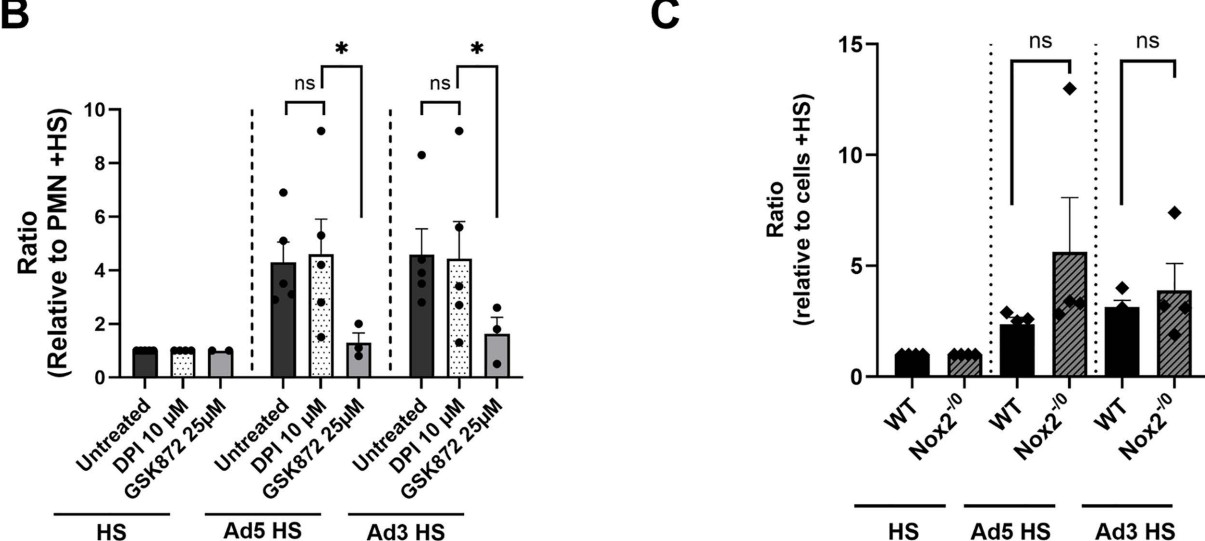

**Fig 8. Ad5- and Ad3-induced RIPK3-dependent and ROS-independent CXCL8 release. (A)** PMNs were incubated with medium (black bar), human serum (HS, gray bar) or HS-opsonized Ads (MOI $10^4$ vp/cell) (Ad5: hatched bar; Ad3: white bar). ELISA was performed on supernatants harvested at 1h or 3h post-incubation. The histograms represent the mean + SEM of CXCL8 concentration in the supernatants. Six independent experiments were performed with technical duplicates for each condition in every experiment. Each dot represents the mean value of the technical replicates for each experiment. Kruskal Wallis test followed by Dunn's multiple comparison tests (comparison with HS condition) were performed. *, $p < 0.05$; **, $p < 0.01$; ***, $p < 0.005$. **(B)** The same experiment as in (A) was performed with PMNs treated or not with RIPK3 kinase (GSK872) or NADPH oxidase NOX2 (DPI) inhibitors. The histograms show the ratio of CXCL8 concentration + SEM for each condition relative to that obtained with HS, after 3h incubation (3 independent experiments for GSK872 and 4 for DPI were performed with technical duplicates for each condition in every experiment). Mann-Whitney test was performed (comparison with HS; ns, non-significant; *, $p < 0.05$). **(C)** WT and NOX2-deficient (Nox2$^{-/0}$) PLB-985 cells were incubated with medium, HS or HS-opsonized virus (MOI $10^4$ vp/cell). CXCL8 concentration in supernatants harvested after 3h incubation was measured by ELISA. The histograms represent the ratios of CXCL8 concentration (mean + SEM) in each condition relative to cells incubated with HS (four independent experiments were conducted with technical duplicates for each condition in every experiment). Mann-Whitney test was performed (comparison with HS). ns, non-significant.

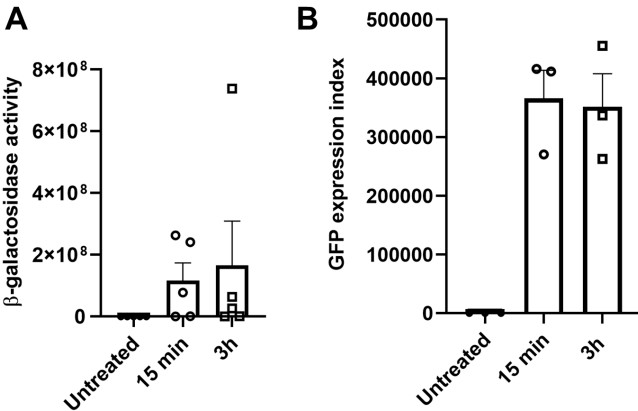

**Fig 9. Ad5 and Ad3 are still infectious after internalization by PMNs.** PMNs were first incubated with HS-opsonized Ad5 **(A)** or Ad3 **(B)** at 4°C then after washing, PMNs were incubated at 37°C for either 15 min or for 3h and harvested. Afterwards, PMN lysates were prepared and added to Ad-sensitive cells. After 20h, β-galactosidase activity (**A**, Ad5) and GFP expression index (**B**, Ad3) were measured to assess the presence of functional Ads in PMN lysates. Means + SEM are shown (**A**, Three independent experiments with technical duplicates for each condition in every experiment; **B**, Three independent experiments).

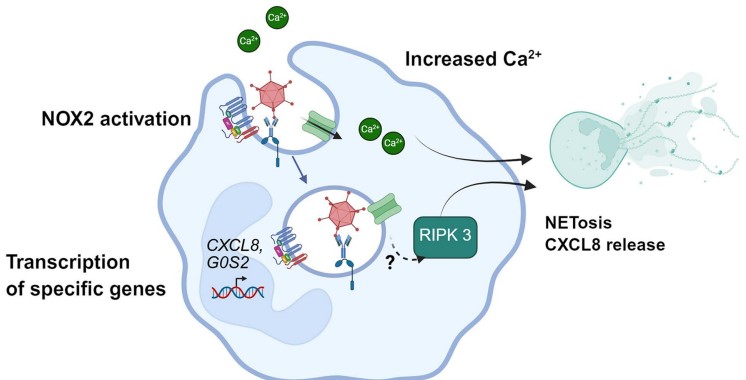

**Fig 10. Interactions between PMNs and Ad unveiled in this study.** IgG-coated Ads bind to PMNs via CD32 receptor recognition. This first step triggers Ad internalization in a CD63+Rab5+ phagosome. Simultaneously, an Ad-specific transcriptional program is induced while NOX2 produces ROS, and an extracellular calcium entry occurs via calcium (or cationic) channels. These events lead, after a few hours, to the PMN lytic death with NET emission, and IL-8 release. This cell death depends on extracellular calcium and RIPK3 (Created in BioRender. Dupré, **S.** (2026) https://BioRender.com/v52f952).

had released NETs. Inhibition of PAD4 or neutrophil elastase (NE), two enzymes involved in NET formation and NETosis (*i.e.,* NET-induced cell death), did not significantly affect the cell death observed after Ad internalization. NE has also been described to cleave Gasdermin D which induces NETosis or cell lysis [56]. Cleavage of Gasdermin D by Caspase 4/11 or caspase 1 is also involved in PMN pyroptosis [56]. However, in our study, the inhibition of Gasdermin D by LDC7559 did not lower PMN death. The PMN death induced by HS-opsonized Ads is also caspase-independent but relies on extracellular calcium. Calcium has been involved in NET formation and NETosis. Singh *et al.* [57] demonstrated that cytoplasmic calcium enhancement in PMNs, using the calcium ionophore A23187, activates calpain that cleaves nesprin-1. Nesprin-1 cleavage weakens the nuclear envelope and promotes NET release. Moreover, Vorobjeva *et al.* demonstrated using the same ionophore, that a calcium overload induced the opening of the mitochondrial transition pore (mPTP) and the production of mitochondrial ROS which led to NETosis [58]. Calcium could also activate the $Ca^{2+}$ calmodulin-dependent kinase

II (CaMKII) which triggers phosphorylation of histone deacetylases and their nuclear export. This favors histone acetylation and de-condensation of chromatin, a process which contributes to NET formation [59]. CaMKII can also be phosphorylated and activated by RIPK3. In cardiomyocytes, activation of CaMKII leads to mPTP opening, mitochondrial ROS production and necroptosis [60]. Our data suggest that RIPK3 is indeed involved in HS-opsonized Ad-induced PMN lysis. As it has been demonstrated for methicillin-resistant *Staphylococcus aureus,* the HS-opsonized Ad-induced PMN death is RIPK3-dependent but RIPK1- and MLKL-independent, the two other proteins that trigger necroptosis. *Staphylococcus aureus* internalization in PMNs also led to IL-1β release in a caspase-1- and inflammasome-independent manner [61]. Thus, HS-opsonized Ad-induced PMN death triggers a non-canonical cell death that involves calcium entry and RIPK3 and is accompanied by NET emission. We still don't know how RIPK3 is activated. In HeLa cells overexpressing RIPK3, an interaction of this protein has been observed with Ad5 capsid proteins [62]. However, we could not observe that Ad5 penetrated the cytosol in our TEM experiments. In dendritic cells and macrophages, internalization of IgG-opsonized Ad induced activation of TLR9 and an inflammasome, respectively AIM2 and NLRP3 [10,11,54]. As a result, TNFα and IL1β are released. We did not check for TLR9 activation, however FcγRIIA can activate the MAPK pathway which could trigger NF-kB nuclear translocation and upregulation of CXCL8 expression. Both Ads triggered chemokine CXCL8 release within 1 h of incubation with PMNs, which further increased over 3h. This release was NOX2-independent but RIPK3-dependent. So, the release of CXCL8 is partly due to PMN death. RIPK3 may also be involved in activating *CXCL8* transcription *via* CaMKII which has been involved in NF-kB activation pathway in macrophages [63].

Having examined the consequences of Ad-PMN interaction for PMNs, we also analyzed the consequences for Ads. We observed that PMN-internalized Ads harvested after 3h of phagocytosis keep their full capacity to transduce target cells. The rapid PMN death triggered by Ad internalization may explain this low Ad degradation. All these results support the idea that the PMN-Ad encounter generates a strong pro-inflammatory environment. Thus, over-recruitment of PMNs could generate severe damage to infected tissues. In line with this, a recent study showed that an elevated proportion of PMNs in circulating leukocytes is a risk factor for severe Ad-induced pneumonia in children [64]. Finally, such consequences could also limit the use of high doses of Ad-based therapeutic tools, as an excessive inflammatory response could trigger more risks for the host than benefits, whether for vaccination or anti-cancer therapy.

## Materials and methods

### Cell culture

The human myeloid leukemia wild-type PLB-985 and X0-CGD PLB 985 cell lines [65] were generous gifts from Dr. Marie-José Stasia. Cells were cultured in RPMI 1640 GLUTAMAX medium (Gibco 61870–010) supplemented with 10% fetal bovine serum (FBS) and 1% of antibiotic and antifungal solution (penicillin G 100 U/mL, streptomycin 100 µg/mL, amphotericin B 250 ng/mL -final concentrations-, EuroBio - CABPSA00-0U), at 37°C, 5% $CO_2$. For all experiments, PLB-985 cells were differentiated into neutrophil-like cells for 5 or 6 days by adding 1.25% DMSO twice, with a medium change for the second dose. In ROS production experiments, interferon γ (20 000 units) was added 24 hours before the experiments. 293A and A549 cell lines are human embryonic kidney cells and lung adenocarcinoma cells, respectively. 293A cell line expresses the E1 region of the adenoviral genome, enabling the production of replication-defective Ads. Both 293A and A549 cells were maintained in Dulbecco modified Eagle medium (Gibco, 31966–021) supplemented with 10% FBS and 1% of non-essential amino acids and split twice a week. Mycoplasma contamination testing was performed on the cell lines every six months.

### Neutrophil isolation

PMNs were isolated from citrate phosphate dextrose solution- or EDTA-peripheral blood obtained from healthy donors (Etablissement Français du Sang, Cabanel & Pitié-Salpêtrière, Paris, 13/NECKER/094 & 2022-2026-003 agreements). The healthy donors are volunteers registered in the "Etablissement Français du Sang" database; we do not have any personal information about them. Two methods of purification were used: (i) purification by dextran sedimentation,

Ficoll-Paque Plus gradient (Cytiva, 17144002) and osmotic lysis of remaining erythrocytes and (ii) isolation using immuno-magnetic beads. For the first method, the blood was mixed with an equal volume of 2% Dextran in 0.9% NaCl to sediment erythrocytes. After 45 min at room temperature, the leukocyte-rich supernatant was collected and centrifuged at 400$g$ for 10 min. The cell pellets were resuspended in PBS and centrifuged through a Ficoll-Paque Plus gradient (cell suspension: Ficoll-Paque, ratio of 2:1) at 400$g$ for 30 min at room temperature. The resulting pellets were resuspended in ice-cold distilled $H_2O$ for 30s to lyse the erythrocytes and osmolarity was restored with an equal volume of 1.8% NaCl. The granulocytes were pelleted at 400$g$ for 10 min at 4°C and then were resuspended in ice-cold PBS. For the second method, PMNs were isolated using the MACSxpress Whole Blood Neutrophil Isolation Kit (Miltenyi Biotec, 130-104-434) and then the MACSxpress Erythrocyte Depletion Kit (Miltenyi Biotec, 130-098-196) making it possible to discard the remaining red blood cells. PMNs were used either immediately or were kept alive in conditioned medium as described by Monceaux *et al*. [66] and used within two days after their isolation.

## Viruses

AE18 (hereafter referred to as Ad5) is based on an Ad5 serotype and is deleted in E1 and E3 regions [67]. Its E1 region is replaced by a β−galactosidase (*lacZ* gene) expression cassette. Ad3-gfp (hereafter noted as Ad3) is based on an Ad3 serotype and its E3 region is replaced by a GFP expression cassette [68]. Viruses were amplified in 293A cells (Ad5) and A549 cells (Ad3) and purified from cell lysates by two successive CsCl ultracentrifugation steps and titrated by spectrophotometry [69]. All viruses were stored at −80°C in PBS-7% glycerol. For microscopy & flow cytometry experiments, viruses were labeled with Alexa-488 (AZDye 488 TFP Ester - Fluoroprobes, 1026–1) or with STAR Orange NHS Ester probe (Abberior) as follows. The virus particles (vp) were resuspended in a buffer containing 10% glycerol and 100 mM $NaHCO_3$ in PBS and $2 \times 10^{11}$ vp were loaded on a Zeba Spin Desalting 7K MWCO column (ThermoFisher Scientific). They were then incubated with the fluorescent probes ($11.94 \times 10^{-8}$ mol) for 1 hour at room temperature, loaded on new columns, resuspended in PBS-10% glycerol and stored at -80°C.

## Chemicals

For opsonization steps, human serum (H4522) was obtained from Sigma-Aldrich. Phorbol Myristate Acetate (PMA) (P-8139) and the NADPH oxidase inhibitor Diphenyleneiodonium (DPI) (D2926) were also from Sigma-Aldrich. For Sytox Green cytotoxicity assay, the following inhibitors were used: Akt inhibitor XI (SantaCruz Technology, sc221229), Z-VAD-FMK (InvivoGen, tlrl-vad), LDC7559 (Sigma-Aldrich, SML3214), Sivelestat (Sigma-Aldrich, S7198), GSK872 (MedChemExpress, HY-101872/CS-7609), GSK840 (Clinisciences, HY1040201), GSK 843 (Clinisciences, HY-125402), Necrosulfonamide (Tebubio, 282T6904), Necrostatin 1s (MedChemExpress, HY-14622A), Dasatinib (MedChemExpress, HY-10181), Reparixin (Sigma-Aldrich, SML2655–5MG) and GSK484 (MedChemExpress, HY-100514).

## CXCL8 assay

PMNs ($5 \times 10^5$) were resuspended in 10% FBS RPMI-1640 medium, either alone as a control, or incubated with human serum (HS) or HS-opsonized Ads. HS-opsonized Ads were obtained by mixing $5 \times 10^9$ Ads (in 20µl of RPMI) with 20µl of human serum (multiplicity of infection (MOI) of $10^4$ viral particles (vp) per cell). After 1 and 3 hours of stimulation at 37°C, the amount of released CXCL8 in supernatants of PMNs was determined using the Human CXCL8 ELISA kit (R&DSystems BioTechne, DY208–05 and DY008B), following the supplier's instructions.

## Binding assay

A488-labeled Ads were diluted in Hanks' balanced salt solution (HBSS, Sigma-Aldrich, H6648) alone, or supplemented with 50% human serum (HS), heat-inactivated HS (HI-HS) or IgG-depleted HS. Cells ($2 \times 10^5$) were resuspended in HBSS buffer in Eppendorf tubes and incubated 45 min at 4°C with the different Ad mixtures to allow binding, at a MOI

of $10^4$ vp/cell. Samples were washed in PBS with 1% bovine serum albumin (BSA), then resuspended in HBSS buffer. A488 was excited at 488 nm and the fluorescence emission was assessed at 533/30 nm for 10,000 cells per condition by flow cytometry (BD Biosciences, BDAccuri C6 Plus). Data were analyzed using BDAccuri C6 Plus software. For Fcγ receptor-blocking experiments, the same protocol was used except that cells were pre-incubated for 20 min at 4°C with the following blocking antibodies: anti-CD16 [3G8] clone antibody (ARG42244, ArigoBio) 5 µg/mL and/or anti-CD32 IV.3 clone antibody (BioXCell BE0224) 10 µg/mL, or control isotypes IgG1 (ARG20767, ArigoBio) 5µg/mL and IgG2 (MPC-11 clone, BioXCell BE0086) 10 µg/mL.

## Immunofluorescence

For all immunofluorescence assays, 12mm glass coverslips were incubated 30 min with polylysine 0.01% at 37°C in 24-well plates and washed 3 times with PBS. Then, $5 \times 10^5$ cells were resuspended in HBSS solution and plated on coverslips for 30 min at 37°C. HS-opsonized A-488nm labeled Ads were incubated with PLB-985 cells or PMNs at 4°C for 30 min (MOI $10^4$ vp/cell) to allow Ad binding. These samples were incubated at 37°C for 10 min and then fixed with 4% paraformaldehyde (PFA) for 15 min at room temperature, washed with 1×PBS, and treated with permeabilization buffer (1×PBS, 0.5% BSA and 0.1% Triton-X-100 (TX)) for 10 min at room temperature. Fc receptors were blocked for 1h at 37°C (or overnight at 4°C) with 10% HI-HS in PBS-BSA 1%. Anti-Rab5 (BD Biosciences – 610724, mouse IgG 1:150), anti-CD63 (mouse, SantaCruz Biotechnology sc-5275, 1:100) or anti-p67 (rabbit, Millipore 07–002, 1:100) were added for 1h at 37°C. Secondary anti-rabbit A-568 nm (Life technologies A11036, 1:500) or anti-mouse A-647 nm (Thermofisher A21235, 1:1000) antibodies were then added for 45 min at 37°C. Finally, the coverslips were mounted on glass slides with DAPI-containing Fluoroshield (Sigma-Aldrich, F6057 - 20 ML) mounting medium.

## Phalloidin-labeling assay

PLB-985 cells or PMNs ($5 \times 10^5$) were resuspended in HBSS, then plated on polylysine-coated glass coverslips for 30 min at 37°C. Afterwards, HS-opsonized A-488nm labeled Ads were added on the cells (MOI $10^4$ vp/cell) and incubated at 4°C with or without further incubation at 37°C for 45 or 120 min. The samples were fixed and permeabilized as described above and incubated with A-568 phalloidin (Invitrogen, A12380, 1:500) for 30 min at 37°C. The coverslips were then mounted on glass slides with DAPI-containing Fluoroshield (Sigma-Aldrich, F6057) mounting medium.

## Neutrophil extracellular trap immunofluorescence

The following protocol was adapted from Brinkmann *et al*. [30]. Cells ($5 \times 10^5$) were resuspended in RPMI medium supplemented with 2% BSA then put on polylysine-coated glass coverslips. The plated cells were incubated for 4 h at 37°C with HS-preincubated Star Orange-labeled Ads (MOI $10^4$ vp/cell) or 100 nM Phorbol Myristate Acetate (PMA). Following fixation with 4% PFA and permeabilization, Fc receptors were blocked at 4°C overnight. The samples were next incubated for 1h at 37°C with anti-myeloperoxidase (MPO) primary antibody (SantaCruz Biotechnology sc52707, 1:100) and then labeled with anti-mouse A-647 nm secondary antibody (Thermofisher 21235, 1:1000). DNA was then counterstained with 1 µg/mL Hoechst solution and coverslips were mounted on glass slides with Abberior antifade solid mounting medium.

## Oxyburst BSA assay

PMNs ($5 \times 10^5$) were resuspended in HBSS solution, plated on polylysine-coated glass coverslips for 30 min at 37°C and 200 µg/mL OxyBURST Green H2HFF BSA (Invitrogen, O13291) was added. After 5 min, the plated PMNs were incubated with HS-opsonized Star Orange-labeled Ads for 45 min at 37°C. Following fixation and permeabilization, the coverslips were mounted on glass slides with DAPI-containing mounting medium.

## Microscopy and imaging analysis

All samples were observed with a spinning disk confocal system (Yokogawa CSU-X1-A1, Yokogawa Electric, Japan) and photographed with a Prime 95B sCMOS camera (Photometrics) at the Imagerie-Gif microscopy facility (IbiSA member, granted by « France-BioImaging -ANR-10-INBS-04–01- & Saclay Plant Science -ANR-11-IDEX-0003-02- Labex). The spinning disk system is mounted on a Nikon Eclipse Ti E inverted microscope, equipped with a X100 oil immersion objective and driven by Metamorph software (7.7 version). A series of 14–20 confocal z-planes were collected for each slide position. Image analysis was performed with ImageJ software (1.54g version). A 3D colocalization analysis was achieved with JaCoP (Just Another Colocalization Plugin) plugin. 3D segmentation was performed using IMARIS software.

## Transmission electron microscopy assay

PMNs or PLB-985 cells ($5 \times 10^6$) were resuspended in HBSS solution then HS-opsonized Ad5 (MOI $10^4$ vp/cell) or HS only were added and the samples were incubated 30 min at 4°C. After a washing step, cells were incubated 10 min and/or 1h at 37°C and fixed with 2% PFA and 2.5% glutaraldehyde in 0.1M cacodylate buffer (pH 7.4) overnight at room temperature with gentle agitation. After washing steps with 0.1M cacodylate buffer, cells were post-fixed with 1% osmium and 1.5% potassium ferrocyanide in 0.1M cacodylate buffer for 1h at room temperature. Samples were washed, included in 2% LMP agarose then dehydrated in increasing concentrations of ethanol and embedded in EPON resin. After a polymerization step at 60°C, the samples were cut with an ultramicrotome (EM UC6, Leica Microsystems) and stained with uranyl acetate and lead citrate. The stained samples were observed with a JEOL 1400 microscope (JEOL Ltd, Tokyo, Japan), using an accelerating voltage of 120 kV (Imagerie-Gif microscopy facility).

## Sytox green cell death assay

PMN or PLB-985 cells ($2 \times 10^5$ per well) were resuspended in HBSS and incubated or not with inhibitors for 30 min at 37°C and seeded in triplicate in 96-well plate (precoated with heat-inactivated FBS for PMN experiments). Cells were stimulated with HS-opsonized Ads (MOI $10^4$ vp/cell), 100 nM PMA or 1% Triton-X-100. A non-permeant DNA probe, Sytox Green (Invitrogen, S7020, 5 µM final concentration) was added just before fluorescence measurement. Fluorescence was evaluated at 525 nm with a TECAN Infinite M200 plate reader every 20 min for 12h. Cell death and/or Sytox-positive cell percentages were calculated by normalizing with TX-treated or inhibitor-treated conditions. At least three independent experiments were conducted.

## ROS production assay

ROS production was measured using a luminol-based chemiluminescence assay in white 96-well plates, which were precoated with heat-inactivated FBS. PMN ($2 \times 10^5$) in HBSS, were incubated for 5 min with 10 µM DPI or DMSO (0.1%), then HS-opsonized Ads (MOI $10^4$ vp/cell) or HS were added to the wells. Luminol (Merk, 123072, 50 µg/mL) and HRP (Merck, P6782,20 U/ml final concentration) were put into the wells. Light emission was recorded by the plate reader "Promega Glomax Multi+ detection system" every 2 min for 200 min at room temperature. The area under the curve was calculated to obtain ROS production during 1 or 2h.

## Cytosolic calcium concentration assay

Cells ($1 \times 10^6$) were resuspended in HEPES Buffer Saline (HBS: 135 mM NaCl, 5.9 mM KCl, 1.2 mM $MgCl_2$, 11.6 mM HEPES, 11.5 mM glucose adjusted to pH 7.3 with NaOH) and loaded with 4 µM of the ratiometric probe Indo-1 (Fisher Scientific, I1223) for 45 min at room temperature with gentle agitation. Cells were resuspended in 2 mL HBS with 1mM $Ca^{2+}$ in a quartz cuvette and inserted into a spectrofluorophotometer (Varian Cary Eclipse), maintained at 37°C *via* a circulating water bath. Indo-1 was excited with 360 nm light and 405 and 480 nm emissions were recorded. After 30s

acquisition, HS or HS-opsonized Ads (MOI $10^4$ vp/cell) were added to the cuvette and the fluorescence was measured for 800s. From fluorescence values, cytosolic calcium concentrations were calculated, following the method described before [70]. In some experiments, 2.5mM EDTA was added 4 min before HS or HS-opsonized Ads.

## Post-phagocytosis Ad survival assay

PMNs ($1 \times 10^6$) were resuspended in HBSS then incubated for 30 min at 4°C with HBSS, HS or HS-opsonized Ad5 or Ad3 (MOI $10^4$ vp/cell) in Eppendorf tubes. After a washing step, PMNs were plated on 24-well plates and incubated for 15 min or 3h at 37°C. Afterwards, samples (PMN plus media) were harvested and freeze-thaw cycles were performed to release viral particles. Phagocytosis lysates were then centrifuged 3800 rpm 10 min and supernatants were stored at -20°C. Afterwards, 300 µL of supernatants were added onto $2.5 \times 10^5$ 293A or A549 cells in order to evaluate respectively Ad5 and Ad3 cell transduction. After 1h at 37°C, 2% FBS DMEM was added. β-galactosidase activity or GFP fluorescence were assessed 20h later.

## β-galactosidase activity assay

293A cells were washed with PBS then incubated with a β-galactosidase-specific buffer ($K_2HPO_4$ 91mM, $KH_2PO_4$ 9.2 mM, Triton-X-100 0.05%, DTT 1mM, 1 Protease Inhibitor Cocktail Tablet -Roche, 11 836 170 001, in water) for 30 min on ice. Bradford assay was performed on lysates and β-galactosidase activity was assessed thanks to Luminescent Beta-galactosidase Detection Kit II (Takara, 631712). Luminescence was read with a TECAN F200 Pro plate reader. Results were expressed as relative light units (RLU) per microgram of proteins.

## A549 cell fluorescence assessment upon viral transduction

A549 cells were washed with PBS, harvested after trypsin treatment and then their fluorescence was evaluated by flow cytometry (BDAccuri C6 Plus, BD Biosciences). Data were analyzed with BD Accuri C6 Plus software. Results were expressed as GFP expression index (i.e., fluorescence mean intensity multiplied by GFP-positive cell percentage).

## 3' Single-cell RNA sequencing & RT-qPCR

PMNs ($5 \times 10^5$) per condition were resuspended in HBSS. HBSS, HS or HS-opsonized Ad5 (MOI $10^4$ vp/cell) were incubated with PMNs for 30 min at 4°C. After washing with HBSS, samples were incubated 90 min at 37°C then filtered with 40µm EASY Strainers (Grenier Bio-One, 542 040) to discard aggregates. PMNs were counted with Trypan Blue to assess cell viability (that was around 80%) then transferred in DNA LoBind tubes. RNA sequencing was performed with 10X Genomics Chromium 3' single-cell RNAseq by Gustave Roussy Institute genomics facility. Bioinformatics analysis was performed by the Gustave Roussy Institute Bioinformatics facility (INSERM US23, CMRS UMS 3655). Raw files were converted to Fastq format using bcl2fastq (version 2.20.0.422 from Illumina). Reads quality control was achieved using fastqc (version 0.11.9) and then pseudo-mapped to the Ensembl reference transcriptome v99 corresponding to the *Homo sapiens* GRCH38 build with kallisto (version 0.46.2). After a quality control step that included empty droplets and doublet removal, scRNA sequencing data were processed in R with Seurat package (version 4.0.4) in order to normalized, scaled and select 2000 Highly Variable Genes. Then, a principal component analysis (PCA) was applied; the number of PCA kept for further analysis was around 20 (21, 23 or 25 depending of the sample). Then, Louvain clustering of cells was performed using a resolution parameter of 0.3 in order to generate a Uniform Manifold Approximation and Projection space (UMAP). An automatic annotation of cell types was achieved by SingleR (version 1.6.1) (with fine-tuning step) and ClustifyR (version 1.5.1), using packages built-in references. Marker genes for Louvain clusters were identified through a «one versus others» differential analysis using the Wilcoxon test through the FindAllMarkers function from Seurat, considering only genes with a minimum log fold-change of 0.5 in at least 75% of cells from one of the groups compared,

and FDR-adjusted p-values < 0.05 (Benjaminin-Hochberg method). Datasets were integrated using the Harmony method. Cerebro App (version 1.3.1)in R-studio was used to visualize results from the bioinformatic analysis. Part of the raw data (one technical replicate) and all processed data used in this study have been deposited in the Gene Expression Omnibus (GEO) under accession number GSE325281.

To confirm the results obtained by RNA sequencing, RT-qPCR was performed on *CXCL8*, *GOS2* & *GAPDH* genes. Cells (PLB-985 cell line or PMNs) were resuspended in HBSS and incubated with HBSS, HS or HS-opsonized Ads for 30 min at 4°C. After a washing step, samples were incubated for 1h at 37°C. Cell pellets were harvested and stored at -80°C. Afterwards, mRNA were extracted with CELLDATA RNAStorm Fresh Cell & Tissue RNA Isolation kit (Biotum CD504) and treated with a DNAse (RQ1 RNAse-free DNAse kit, PROMEGA, M6101). RT-qPCR was performed with GoTaq 2 step RT-qPCR kit (PROMEGA, A6010). qPCR primer sequences (Biomol GmbH) for *CXCL8* gene were 5'-TAGCAAAATTGAG GCCAAGG-3' (sense) and 5'AGCAGACTAGGGTTGCCAGA-3' (antisense), and *G0S2* primer sequences were 5'-TCAGAGAAACCGCTGACATC-3' (sense) and 5'-CCTCCCTAGTGCAAAATGGT-3' (antisense). Primers against *GAPDH* gene were used as positive control: 5'-GAGTCAACGGATTTGGTCGT-3' (sense) and 5'-TTGATTTTGGAGGGATCTCG-3' (antisense). q-PCR was performed with StepOnePlus apparatus (Applied Biosystems).

## Statistics analysis and reproducibility

Graphpad prism 9 software (GraphPad Software, USA) was used for the statistical analyses as indicated in the figure legends. Each statistical test is indicated in the legend. The number of replicates is indicated in each figure legend.

## Supporting information

**S1 Fig. IgG-dependent Ad5 and Ad3 binding to PLB-985 cells. (A)** Alexa-488 labeled Ad5 **(A)** or Ad3 **(B)** were incubated at 37°C with medium, human serum (HS), heat-inactivated (HI-HS) or IgG-depleted HS. The PLB-985 cells were then exposed to Ads at 4°C to allow cell binding (MOI $10^4$ vp/cell). The results are presented as an association index + SEM (*i.e.,* mean fluorescence intensity multiplied by percentage of positive cells). Each dot represents the association index for one experiment. A Kruskal-Wallis test was performed followed by Dunn's multiple comparison tests. to compare binding in the HI-HS and IgG-depleted HS conditions relative to the HS condition. *, p < 0.05; **, p < 0.01; ***, p < 0.001.
(TIF)

**S2 Fig. Presence of anti-Ad antibodies in the human serum.** Titers of anti-Ad IgGs in the human serum used for opsonization were determined by ELISA. A Mann-Whitney test showed that there was no significant difference (ns) between the Ad5 and Ad3 titers, expressed as the mean ± SEM on a $\log_2$ scale. Three independent experiments were performed.
(TIF)

**S3 Fig. Mean fluorescence and percentage of Ad-positive cells for PMNs incubated with HS-opsonized Ads.** A-488 labeled Ad5 or Ad3 were incubated at 37°C with human serum (HS). PMNs were then exposed to these Ads at 4°C to allow cell binding (MOI $10^4$ vp/cell). **(A)** Percentage of Ad-positive cells + SEM, indicating the number of cells with Ads at their surface; **(B)** Mean fluorescence representing the amount of Ad bound per cell; **(C)** Association index calculated from the previous data. Welch's t test was performed. ns, non-significant; *, p < 0.05. Seven independent experiments were performed.
(TIF)

**S4 Fig. Ad5 is internalized in phagosomes in PLB-985 cells.** PLB-985 cells were incubated with human serum-opsonized A-488 Ad5 (MOI $10^4$ vp/cell) for 1h at 37°C, then prepared for electronic microscopy. The images show an Ad5-infected cell with a phagosome engulfed viral particles (inset).
(TIF)

**S5 Fig. NOX2-deficient PLB-985 cells do not produce ROS.** WT & NOX2$^{-/0}$ PLB-985 cells were incubated with buffer or human serum-opsonized Ad5 (MOI 10$^4$ vp/cell). Analysis of ROS production over 1 hour, detected by a luminometry-based test, showed that in WT cells, Ad5 infection resulted in significantly increased ROS production but that in NOX2-deficient PLB-985 cells ROS production did not increase following infection, and indeed remained very low. An Anova-test followed by t-tests with Welch's correction was performed: ns, non significant; *, $p < 0.05$; **, $p < 0.01$. Four independent experiments were conducted with technical duplicates for each condition in every experiment.
(TIF)

**S6 Fig. Ads induce Ca$^{2+}$ increase in PLB-985 cells Cytosolic Ca$^{2+}$ concentration ([Ca2$^+$]$_{cyt}$) measurement in PLB-985 cells using Indo-1 fluorescence.** Cells were first preincubated with either buffer alone or with buffer + EDTA 2.5 mM for 5 min. Then, after 30 s control measurement, either human serum (HS) (green trace), or HS-opsonized Ad3 (red trace) (**A**) or Ad5 (black and blue trace) (**A,B**) were added (black arrow). In (**B**), cells were pretreated with 2.5 mM EDTA or not, and exposed to HS-opsonized Ad5 (untreated cells: black curve, EDTA treatment: blue). Results are representative of 3 independent experiments.
(TIF)

**S7 Fig. Calcium-independent Ad5 internalization. (A)** PLB-985 cells were exposed to human serum-opsonized A-488 Ad5 (MOI 10$^4$ vp/cell) for 30 min at 4°C to allow binding then incubated for 0 or 45 min at 37°C. The left panel shows untreated cells while in the right panel, cells were pre-incubated with 2.5 mM EDTA. Extracellular A-488 Ads accessible to A-594 anti-IgG antibody are labeled in yellow (A-488-labeled and A-594 anti-IgG-labeled double positive particles) while intracellular Ads are only A-488-positive and appeared as green particles. The plasma membrane was stained with WGA CF-640R (gray) and nuclei were stained with DAPI (blue). **(B)** Quantification of the percentage ± SEM of internalized Ad patches after 45 min of incubation (3 independent experiments, n = 25 cells for the untreated condition, n = 24 for the condition with EDTA). Each dot represents the percentage of internalized Ad patches for each experiment. Mann-Whitney test was performed. ns, non significant.
(TIF)

**S8 Fig. Single-cell RNA sequencing of blood-purified PMNs reveals several subpopulations. (A)** Single-cell RNA-sequencing analysis of purified blood PMNs and incubated with either human serum (HS) or with HS-opsonized Ad5 for 1 h. Identification of 7 cell clusters using the SEURAT package. The clusters are visualized using Uniform Manifold Approximation and Projection (UMAP). Each dot represents one cell. **(B)** Upper panel: Log normalized expression level of *NAMPT, CXCR2, SOD2, CSF3R, FCGR3B* in the different clusters. Lower panel: as above for the interferon stimulated genes: *STAT1, GRB2, IRF1.*
(TIF)

**S9 Fig. Increased expression of *CXCL8* and *G0S2* in the presence of Ad5.** After normalization for *GAPDH* gene expression, the histogram shows the relative expression of *CXCL8* and *G0S2* mRNAs in PMNs in the presence of Ad5 as a function of the expression in cells incubated with HS alone. Data show the mean ± SEM and are from three experiments with technical duplicates for each condition in every experiment.
(TIF)

**S10 Fig. Ad5 are trapped in NETs.** PMN were exposed or not to human serum-opsonized A-488 Ad5 (MOI 10$^4$ vp/cell) for 30 min at 4°C to allow binding then centrifuged and washed to eliminate unbound Ads. The cells were then incubated for 3 hours at 37°C. The upper panel shows PMNs alone and the lower panels PMNs incubated with HS-opzonized A-488 Ads (green). The plasma membrane was stained with WGA CF-640R (magenta) and nuclei were stained with DAPI (blue). **(A)** Representative images acquired by spinning disk confocal microscopy. Each image represents a projection of a Z

stack. Scale bar, 5µm **(B)** 3D segmentation of the merge images of A (lower panels) to highlight NETs and Ad5 trapped in NETs. Scale bars, 5µm.
(TIF)

**S11 Fig. Ad5 and Ad3 induce cell death associated with NET production in PLB-985 cells. (A)** Ad cytotoxicity assessed via Sytox Green assay. PLB-985 cells were incubated with HBSS, Triton-X-100 1%, or HS-opsonized Ads (MOI $10^4$ vp/cell) in presence of Sytox Green probe, a cell-impermeant DNA fluorescent probe. Cell fluorescence was measured with a plate reader. The curves represent the cell death percentage (mean±SEM, *i.e.,* cell fluorescence relative to Triton-X-100-lysed cell fluorescence multiplied by 100) at different times post-infection.Mann-Whitney test was performed. *, $p<0.05$; **, $p<0.01$; (comparison with PLB-985 cells incubated with HBSS). Five independent experiments, with technical triplicates for each condition in every experiment, were performed. **(B)** Ad-induced NET (neutrophil extracellular trap) formation. PLB-985 cells were incubated with buffer (Mock), PMA 100 nM (positive control) or labeled and HS-opsonized Ad5 or Ad3 (MOI $10^4$ vp/cell) for 4h, then fixed, permeabilized and stained for DNA (blue) and myeloperoxidase (MPO). Scale bar=10 µm. Five independent experiments, with technical triplicates for each condition in every experiment, were conducted.
(TIF)

**S12 Fig. Ads induce PLB-985 WT and NOX2-deficient cell death. (A)** PLB-985 WT cells were incubated with Sytox Green probe, then with HS (blue), HS-opsonized Ad5 (green) or PMA 100 nM (red). Sytox Green fluorescence was measured at the indicated time points and expressed relative to the cells treated with Triton-X-100 (+ SEM). **(B)** The same experiment was performed on NOX2-deficient (NOX2$^{-/0}$) PLB-985 cells. Six (A) and four (B) independent experiments, with technical triplicates for each condition in every experiment, were performed.
(TIF)

**S13 Fig. RIPK3- and calcium-dependent Ad3-induced cell death.** The same experiment as shown in Fig 7D was performed with Ad3. PMNs were pre-incubated with different inhibitors in the presence of Sytox Green probe and incubated with HS-opsonized Ad3. After 4h, Sytox Green fluorescence was measured and expressed as a ratio relative to the fluorescence of the Ad-exposed cells treated with the respective inhibitor or HS. The condition without inhibitors (untreated cells) was set at 100%. Kruskal-Wallis test was performed followed by Dunn's multiple comparison tests: **, $p<0.01$. Three independent experiments, except for EDTA condition (2 experiments), with three technical replicates per condition for each experiment were performed.
(TIF)

**S1 File. Supplementary file: Supplementary material and methods.**
(DOCX)

**S2 File. Supplementary file: Data used in the graphs of all the figures and supplementary figures.**
(XLSX)

## Acknowledgments

We would like to thank Dr. Kirsty Grant for her assistance with the English correction and for critically reviewing the manuscript. The facilities used in this study were funded by the Institut Gustave Roussy (https://www.gustaveroussy.fr; NTA2022) and the Agence Nationale de la Recherche (https://anr.fr; ANR-10-INBS-04/FranceBioImaging; ANR-11-IDEX-0003-02/ Saclay Plant Sciences).

## Author contributions

**Conceptualization:** Salomé Laurans, Karim Benihoud, Sophie Dupré-Crochet.

**Data curation:** Salomé Laurans, Rémy Jelin, Sophie Dupré-Crochet.

**Formal analysis:** Salomé Laurans, Rémy Jelin, Sophie Dupré-Crochet.

**Funding acquisition:** Salomé Laurans, Karim Benihoud, Sophie Dupré-Crochet.

**Investigation:** Salomé Laurans, Soizic Huerre, Olivier Dellis, Céline Férard, Hadrien Jalaber, Clément Vanbergue, Emilie Brun, Sophie Dupré-Crochet.

**Methodology:** Salomé Laurans, Soizic Huerre, Céline Férard, Emilie Brun, Sophie Dupré-Crochet.

**Project administration:** Sophie Dupré-Crochet.

**Resources:** Oliver Nüsse, Karim Benihoud, Sophie Dupré-Crochet.

**Supervision:** Karim Benihoud, Sophie Dupré-Crochet.

**Validation:** Salomé Laurans, Sophie Dupré-Crochet.

**Visualization:** Salomé Laurans, Sophie Dupré-Crochet.

**Writing – original draft:** Salomé Laurans, Sophie Dupré-Crochet.

**Writing – review & editing:** Salomé Laurans, Soizic Huerre, Olivier Dellis, Emilie Brun, Oliver Nüsse, Karim Benihoud.

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
