## [Decision Letter · Decision Letter 0]

18 Sep 2025

Adenovirus phagocytosis by neutrophils triggers a pro-inflammatory response

PLOS Pathogens

Dear Dr. Dupré-Crochet,

Thank you for submitting your manuscript to PLOS Pathogens. After careful consideration, we feel that it has merit but does not fully meet PLOS Pathogens’s publication criteria as it currently stands. Therefore, we invite you to submit a significantly revised version of the manuscript that addresses the points raised during the review process by reviewers 2 and 3. Please pay particular attention to the points raised by Reviewer 2 about submitting higher quality and more compelling microscopy images. Please also address the significant critiques raised by Reviewer 3 about both the technical and conceptual novelty of your study.

Please submit your revised manuscript within 60 days Nov 17 2025 11:59PM. If you will need more time than this to complete your revisions, please reply to this message or contact the journal office at plospathogens@plos.org. When you’re ready to submit your revision, log on to https://www.editorialmanager.com/ppathogens/ and select the ‘Submissions Needing Revision’ folder to locate your manuscript file.

* A rebuttal letter that responds to each point raised by the editor and reviewer(s). You should upload this letter as a separate file labeled ‘Response to Reviewers’. This file does not need to include responses to any formatting updates and technical items listed in the ‘Journal Requirements’ section below.

* A marked-up copy of your manuscript that highlights changes made to the original version. You should upload this as a separate file labeled ‘Revised Manuscript with Track Changes’.

* An unmarked version of your revised paper without tracked changes. You should upload this as a separate file labeled ‘Manuscript’.

We look forward to receiving your revised manuscript.

Kind regards,

Kinjal Majumder, PhD

Guest Editor

PLOS Pathogens

Blossom Damania

Section Editor

PLOS Pathogens

Editor-in-Chief

PLOS Pathogens

orcid.org/0000-0003-2946-9497

Editor-in-Chief

PLOS Pathogens

orcid.org/0000-0002-7699-2064

**Journal Requirements:**

2) We noticed that you used the phrase ‘data not shown’ in the manuscript. We do not allow these references, as the PLOS data access policy requires that all data be either published with the manuscript or made available in a publicly accessible database. Please amend the supplementary material to include the referenced data or remove the references.

- ® on page: 19.

- TM on page: 23.

**Reviewers’ Comments:**

Reviewer’s Responses to Questions

**Part I - Summary**

Reviewer #1: Laurans et al revisit human adenovirus phagocytosis by neutrophils. The last paper in this area seems to have been published in 2005 and the progress in tools and reagents over the last 20 years certainly justifies a revisit with new technology.

The authors use fluorescently labelled Ad3 or Ad5 to infect primary human neutrophils or PLB-985 cells at an MOI of 10,000 (same MOI as Cotter et al in 2005). Similar to the former study, this study confirms that Ad association with neutrophils requires IgG recognition and CD32 and results in internalization. Using confocal microscopy and TEM, they confirm that virus is located in membrane delimited compartments that colocalize with Rab5 and CD63, markers of the phagosome and triggers NOX2 dependent ROS production that is dependent on extracellular calcium.

The novelty of the current submission is related to single cell RNA sequencing reveals that Ad interaction triggers a unique transcriptional response, including upregulation of CXCL8. In addition, this study goes on to investigate the outcome of Ad interaction with PMNs, showing that exposure to Ad triggers rapid PMN death, that is RIPK3 dependent but not ROS dependent, and releases NETs and CXCL8. Thus, Ad induced PMN death appears related to “necroptosis”. Additional data, suggests that phagocytosed Ad were not killed and remain functionally able to infect target susceptible cells.

Overall, I thought this was a nice extension of the Cotter paper from 2005 that builds on those observations to show that Ad engulfment by PMNs leads to changes in transcriptional programs and rapid killing of PMNs. This work also begins to parse out the mechanism of how this is achieved.

Reviewer #2: Comments to the submission by Laurans et al. ,Adenovirus (AdV) phagocytosis by neutrophils triggers a proinflammatory response.’ Immune cell interactions with viruses drive both anti-viral defense and pathogenesis. In this context, AdV neutrophil interactions have not been systematically explored, and much is to be discovered. AdV are pathogens of vertebrates, and in humans they cause a variety of disorders ranging from respiratory and gastrointestinal tract infections to acute viremia with systemic effects, often followed by viral persistence for years. Previous studies of AdV interactions with transformed and nontransformed cell lines and primary cells, including dendritic cells, natural killer cells and macrophages, set the stage for the type of work presented here with neutrophiles.

The authors apply conventional cytometry, bulk RNAseq and some single cell light and electron microscopy to describe interactions of two human AdVs, AdV-C5 and AdV-B3 with human neurophils. Their results show that polymorphonuclear neutrophils (PMNs) or neutrophile-like cells undergo cell death within a few hours after inoculation with immunocomplexed Ad5 or Ad3. Unfortunately, the study is descriptive, and does not provide molecualr mechanisms for AdV triggered neutrophile death.

General remarks

In the current version of this mansucript, reference to the literature is not sufficient, and the critical points brought up in introduction and discussion sections are sparsely supported by references and should be enriched with more contextual information. For example, AdV interactions with immune cells and the cell response thereupon, including virus cell entry and gene delivery as well as innate detection by host DNA sensors along with cytokine signalling have been described before in primary cells as well as cell lines, such as THP1. Notably, it had been shown earlier that AdV targeted to Fc receptors penetrates to the cytosol and delivers genes to the cell nucleus, along with being sensed by cGAS triggering the expression of a range of cytokines and chemokines. It would be important to crossreference to the data from previous reports on AdV interactions with other immune cells, and also discuss why there is no AdV present in the cytosol of the PMNs and PLB-985 cells. This seems to be critical to block viral gene expression and antagonism of neutrophile cell death.

Recommendation

To reach par with previous studies on AdV host cell interactions and allow for comparision of AdV infection of other cell types based on previous studies the authors should better characterize AdV induced neutrophil death, Ca influx, CXCL8 release and especially also the neutrophil extracellular trap (NET) formation process. This would potentially make this study an important and valuable contribution. Besides small chemical inhibitors used already, further experiments should include perturbations with RNA interference, use of virus mutants impaired in gene delivery, UV inactivated inocula as well as critical dose-response experiments to assess specificity and sensitivity. Further details see below.

Reviewer #3: The manuscript by Laurans et al examines the interaction of adenoviruses (Ad3, Ad5) with neutrophils. The authors examine adenovirus binding to neutrophils followed by phenotypic effects including calcium mobilization, ROS formation, transcriptomics, CXCL8 release, cell death, neutrophil extracellular trap formation, and viral resilience. The study is done completely in vitro and relevance to anti-adenoviral immunity in vivo is unclear. The first part of the paper recapitulates a lot of work performed in the field more than 20 years ago. The newer aspects of the study are not conducted in sufficient depth, and data are not tied together into a unified conceptual framework.

**Part II – Major Issues: Key Experiments Required for Acceptance**

Please use this section to detail the key new experiments or modifications of existing experiments that should be absolutely required to validate study conclusions.required to validate study conclusions.required to validate study conclusions.required to validate study conclusions.

Generally, there should be no more than 3 such required experiments or major modifications for a “Major Revision” recommendation. If more than 3 experiments are necessary to validate the study conclusions, then you are encouraged to recommend “Reject”.

Reviewer #1: 1) I am not sure this is a major issue. However, I find it hard to reconcile how infection at the incredibly high MOI of 10,000 (same as Cotter used), which leads to over 50% of cells associated with Ad and also induces 50% of cells to die by 4 hrs post exposure, lead to cluster 6 (those associated with Ad) being only 1% or less of the cells on the UMAP shown in Fig. S8. Some explanation should be provided for this disconnect, which seems like a major inconsistency.

2) This is really a minor issue, but not as minor as those listed below. It would be nice to know how far the pH changes in the endosome during Ad internalization before PMN death. Given that the virus remains infectious, it can’t drop far enough to cause capsid disassembly. Is there some interference with acidification? Some discussion on this point would be warranted, if not an experiment.

3) I find the final section on the ability of the Ad to remain infectious after phagocytosis and PMN death to be very interesting, but not well developed. In theory, at least some of the IgG in pooled human serum would be expected to be “neutralizing” and block infection. The fact that Ad that has passage through PMN phagocytosis remains infectious is very interesting and a potentially exciting observation. Is it possible that the phagocytosis process removes neutralizing antibody, actually reactivating IgG neutralized virus? This would be a VERY good trick for a virus, but a new assay would have to be set up to see if infectious virus could be recovered from fully IgG neutralized virus by passage through PMNs. Some discussion on this point would be warranted, if not an actual experiment.

Reviewer #2: Specific points

1) The quality of the immunofluorescence images is generally low. Representation of these data needs to be improved throughout the study. For example, even the DAPI signals of control cells appear to be out of focus (see, e.g., Fig. 2, 3, 4, and in part Fig. 6).

2) The annotation of the figures, including images is insufficient. The composites are not self explanatory and should be improved.

3) Line 95: Ref 18 is for Ad7 not Ad3. Please amend.

4) Fig 1A: What is an association index? I guess these are some normalized fluorescence units of virus particles, but it is unclear if these signals are composed of virus aggregates or monodispersed particles. Please provide fluorescence images.

5) Line 128: Another possibility is a different amount of monodispersity of the inocula. To control for the quality of the fluorescently labeled Ad5 and Ad3 inocula, the authors should run scatter analyses or fluorescence imaging of the labeled virions.

6) Line 130: please check / comment on the presence / absence of the high affinity Fc receptor CD64 in the cells used in this study. If the high affinity CD64 Fc receptor is detected by flow cytometry, e.g., then its effects on Ad-immunocomplexes should be tested, given that anti-CD32 and anti-CD16 antibodies had only small effects on cell association of the immunocomplexes.

7) Fig. 2: It seems as if only a minority of cells binds immunocomplexed Ad3 or Ad5. It is not clear how representative these data are. Please provide better resolved images, and quantifications of single cell analyses.

8) What is the mechanism of cell death of the neutrophile-like cells? Why does the NADPH oxidase (NOX2) inhibitor DPI inhibit cell death triggered by PMA but not immunocomplexed AdV?

9) There is a substantial risk for off-target effects of DPI, including cell tox. What was the minimal concentration of DPI necessary to inhibit ROS production and cell death? Please provide dose-response curves here, at least for the PMNs and PLB-985 cells. The latter lack NOX2 and should not be subject to on-target effects of DPI.

10) Line 171: data not shown. Please provide the data.

11) Line 173: Cells treated with the Ca chelator EGTA did not exhibit Ca increase in the cytosol upon application of immunocomplexed AdV. The authors interprete this to mean that Ca is not released from internal stores in AdV inoculated cells, also in absence of EGTA. This stands on weak grounds. How can the authors exclude the possibility that Ca chelation depletes the internal Ca stores that would normally respond to immunocomplexed AdV?

12) A related follow-up question is: What is the mechanism of Ca influx in these experiments? There is evidence in the literature that AdV entry into cells triggers Ca transients involving extracellular Ca from the medium via pores in the plasma membrane. Is there a similar mechanism at play here?

13) Fig. 5, regarding transcriptome. It is interesting to note that the genes G0S2 and CXCL8 were upregulated in cells treated with immunocomplexed AdV. Unfortunately, these observations are not followed up. Please amend.

14) Fig. 6B: The images available to this reviewer are not of sufficient quality to unequivocally distinguish NETs from nuclei. Can the authors provide nuclear envelope marker stainings to distinguish the bona fide nuclei from the NETs?

15) Fig. 7B: GSK872 used at 25 uM inhibited cell death. At such high concentrations, GSK872 will have off-target effects. Other RIPK3 inhibitors along with dose response curves and RNAi should be used to critically assess the GSK872 results.

16) Fig. 8A: missing annotation of the different bars.

17) Line 260: The inhibition of the CXCL8 receptor by reparixin did not inhibit PLB-985 cell death triggered by AdV. Please amend.

18) Line 503: this must be a wrong reference. Please correct.

Reviewer #3: 1) The paper lacks focus and context. What is the importance of neutrophils and antiviral immunity? What is the central premise of the paper? Adenovirus binding to neutrophils? Adenovirus-induced neutrophil death? Adenovirus induced NET formation? The study touches on all of these subjects, but only superficially and the data are not tied together to create conceptual unity. Thus no real conclusions can be drawn.

2) The adenovirus binding studies are done at 4 degrees C. The relevance of these findings is questionable. Studies should also be performed at 37 degrees to determine if viral internalization is also impacted by other cell receptors and pathways.

3) The human serum used in the study was purchased from Sigma, and from male donors. Experiments should be performed with fresh serum/plasma from both male and female donors.

4) The relevance of the transcriptome studies is questionable. First, it does not make a lot of sense to do single cell transcriptomics on a homogenous population of neutrophils in vitro. Second, neutrophil effector functions are not normally transcriptionally regulated, and many of the genes (aside from CXCL8) have no relevance to the central premise and other observations of the paper. For example, what is the point of highlighting GOS2, a gene possibly involved in apoptosis when a) early apoptosis is generally a post-translational event and b) the authors show that neutrophils do not predominantly die by apoptosis. Furthermore, the transcriptome experiment was done at 90 minutes, barely enough time for all transcription events to occur.

5) The cell death inhibitor studies lack focus and are difficult to interpret. The major effect is observed by the necroptosis inhibitor GSK872 and EDTA, but never linked to the other data in the paper. What is the relationship between NETs and RIPK3? NETs and Ca+2, why was there no cell death by NETosis? Additional studies should be performed to determine if RIPK3 and Ca+2 inhibition also prevent neutrophil NET formation. What is the relationship of all of this to the transcriptomic data?

6) There are off target effects for all the inhibitors listed and more specific inhibition approaches should be utilized to validate findings (eg. z-VAD blocks all caspases, apoptosis and pyroptosis, EDTA chelates all divalent cations and could have other effects on cellular membranes).

7) The re-infection/transduction study performed is artificial and lacks relevance. In an in vivo infection, where NETosis does not occur in isolation, would adenoviruses released from dying?NET neutrophils still be infectious?

**Part III – Minor Issues: Editorial and Data Presentation Modifications**

Reviewer #1: Minor points:

1) Is it really accurate to say that Ad3 and Ad5 use different intracellular trafficking (line 95)? Not sure that is what ref. 18 concludes.

2) Probably need at least one paragraph break between lines 207-241.

3) Don’t use the word ignore on line 373. You are not ignoring the mechanism, it just remains unknown at this time.

4) The method for recovery of virus from PMNs is not clear. On line 578 it is not clear if the harvested samples are total cells+media or just cell pellets, which are then freeze thawed. Please clarify.

5) Minor note: “Lore” indicates that purified Ad aggregated unless they are suspended in buffer containing Mg and Ca cations. This may even be stated in the 2005 Cotter paper (ref 17). This group seems to use basic PBS, without supplementation with Ca2+ and Mg2+, which can reduce viral sample homogeneity and experimental reproducibility. This is not an issue in serum or regular tissue culture medium, which contain both divalent cations.

Reviewer #2: see above

Reviewer #3: The graphs should show individual data points.

PLOS authors have the option to publish the peer review history of their article (what does this mean?). If published, this will include your full peer review and any attached files.). If published, this will include your full peer review and any attached files.). If published, this will include your full peer review and any attached files.). If published, this will include your full peer review and any attached files.

...

Reviewer #1: No

Reviewer #2: No

Reviewer #3: No

**Figure resubmission:**

**Reproducibility:**



---

## [Editor Report · Decision Letter 1]

23 Mar 2026

Dear Pr Dupré-Crochet,

We are pleased to inform you that your manuscript ‘Adenovirus phagocytosis by neutrophils triggers a pro-inflammatory response’ has been provisionally accepted for publication in PLOS Pathogens.

Should you, your institution’s press office or the journal office choose to press release your paper, you will automatically be opted out of early publication. We ask that you notify us now if you or your institution is planning to press release the article. All press must be co-ordinated with PLOS.

Best regards,

Kinjal Majumder, PhD

Guest Editor

PLOS Pathogens

Blossom Damania

Section Editor

PLOS Pathogens

Sumita Bhaduri-McIntosh

Editor-in-Chief

PLOS Pathogens

orcid.org/0000-0003-2946-9497

Michael Malim

Editor-in-Chief

PLOS Pathogens

orcid.org/0000-0002-7699-2064
---

## [Editor Report · Acceptance letter]

Dear Pr Dupré-Crochet,

We are delighted to inform you that your manuscript, “Adenovirus phagocytosis by neutrophils triggers a pro-inflammatory response,” has been formally accepted for publication in PLOS Pathogens.

Soon after your final files are uploaded, the early version of your manuscript, if you opted to have an early version of your article, will be published online. The date of the early version will be your article’s publication date. The final article will be published to the same URL, and all versions of the paper will be accessible to readers.

Best regards,

Sumita Bhaduri-McIntosh

Editor-in-Chief

PLOS Pathogens

orcid.org/0000-0003-2946-9497

Michael Malim

Editor-in-Chief

PLOS Pathogens

orcid.org/0000-0002-7699-2064